# Phenotypic diversification by enhanced genome restructuring after induction of multiple DNA double-strand breaks

Nobuhiko Muramoto[1], Arisa Oda[2,3], Hidenori Tanaka [1], Takahiro Nakamura[2], Kazuto Kugou [2,6]
Kazuki Suda[2], Aki Kobayashi[2], Shiori Yoneda[2], Akinori Ikeuchi[1], Hiroki Sugimoto[1], Satoshi Kondo[4],
Chikara Ohto[4], Takehiko Shibata[5], Norihiro Mitsukawa[1] & Kunihiro Ohta[2,3]

DNA double-strand break (DSB)-mediated genome rearrangements are assumed to provide diverse raw genetic materials enabling accelerated adaptive evolution; however, it remains unclear about the consequences of massive simultaneous DSB formation in cells and their resulting phenotypic impact. Here, we establish an artificial genome-restructuring technology by conditionally introducing multiple genomic DSBs in vivo using a temperature-dependent endonuclease TaqI. Application in yeast and *Arabidopsis thaliana* generates strains with phenotypes, including improved ethanol production from xylose at higher temperature and increased plant biomass, that are stably inherited to offspring after multiple passages. High-throughput genome resequencing revealed that these strains harbor diverse rearrangements, including copy number variations, translocations in retrotransposons, and direct end-joinings at TaqI-cleavage sites. Furthermore, large-scale rearrangements occur frequently in diploid yeasts (28.1%) and tetraploid plants (46.3%), whereas haploid yeasts and diploid plants undergo minimal rearrangement. This genome-restructuring system (TAQing system) will enable rapid genome breeding and aid genome-evolution studies.

[1] Genome Engineering Program, Strategic Innovative Research-Domain, Toyota Central R&D Laboratories, Inc., Nagakute, Aichi 480-1192, Japan. [2] Department of Life Sciences, Graduate School of Arts and Sciences, The University of Tokyo, Komaba 3-8-1, Tokyo 153-8902, Japan. [3] Universal Biology Institute, The University of Tokyo, Hongo 7-3-1, Tokyo 113-0033, Japan. [4] Future Project Division, Toyota Motor Corporation, 1 Toyota-cho, Toyota 471-8572, Japan. [5] Cellular & Molecular Biology Laboratory, RIKEN Advanced Science Institute, Wako-shi, Saitama 351-0198, Japan. [6] Present address: Department of Frontier Research, Kazusa DNA Research Institute, Kisarazu, Chiba 292-0818, Japan. These authors contributed equally: Nobuhiko Muramoto, Arisa Oda, Hidenori Tanaka, Takahiro Nakamura. Correspondence and requests for materials should be addressed to A.O. (email: odar@bio.c.u-tokyo.ac.jp) or to N.M. (email: e1190@mosk.tytlabs.co.jp) or to K.O. (email: kohta@bio.c.u-tokyo.ac.jp)

Synteny analyses of current genomes demonstrate that genome evolution accompanies vestiges of massive genome rearrangements, such as translocations (TLs), deletions, insertions, and copy number variations (CNVs)[1–4]. Human cancer cells also undergo multiple genome rearrangements[5], analogous to those occurring during genome evolution and that might promote their acquisition of malignancy. It is postulated that the source of those rearrangements is DNA double-strand breaks (DSBs) in living cells induced by ionizing radiation, oxidative DNA damages, chemical modification, or abortive DNA replication; however, the causal relationship between formation of multiple DSBs and genome rearrangements requires in-depth investigation.

Genome rearrangements are thought to increase phenotypic diversity that provides critical raw materials for evolution. Point mutations represent another source of raw genetic materials for natural selection, but provide relatively small changes in DNA sequences and enable rather limited exploration of the sequence space via "fitness–random walk" for adaptation to given environmental changes. On the other hand, gross genome rearrangements, such as TLs, CNVs, and aneuploidy, might allow more extensive changes to the sequence space, thereby enabling rapid and comprehensive searches for more favorable DNA sequences. However, much remains to be elucidated about the real impact of genome rearrangements on phenotypic diversification.

Another important event during genome evolution associated with natural selection is "whole-genome duplication (WGD)"[6–8], which is reportedly induced immediately following the large-scale genome rearrangements. For example, the yeast Saccharomyces cerevisiae arose from an ancient WGD along with subsequent genome rearrangements, and Arabidopsis thaliana and various cereal crops evolved through multiple rounds of WGD accompanied by genome rearrangements[6–12]. In addition, it has been demonstrated that polyploids have more evolutionary space for genome rearrangements and these have a major impact on phenotypes in Brassica napus[11,12] and in S. cerevisiare[13]. Genome rearrangement and WGD might contribute synergistically to genome evolution, but it is difficult to assess the relative impact of DSB induction and genome rearrangements in diploid vs haploid yeast, or tetraploid compared to diploid plants.

To address these questions, we establish a new tool to induce artificial large-scale genome rearrangements by introducing multiple genomic DSBs, which are formed simultaneously in living cells by the heat-activated endonuclease TaqI from Thermus thermophilus HB8[14,15]. By applying this system to S. cerevisiae and A. thaliana cells exhibiting different ploidy, we successfully monitor the process of genome rearrangements after formation of multiple DSBs, as well as the impact of each rearrangement on phenotypic diversification.

Notably, analyses of long-read/short-read high-throughput resequencing and chromosomal size reveal that multiple simultaneous DSB formation caused gross genome rearrangements at a higher frequency in diploid yeast cells than in haploid cells. Additionally, induced DSBs promote large-scale genome rearrangements in Arabidopsis more readily in tetraploids than in diploids. Moreover, genome-rearranged yeast cells and plants show enhanced phenotypic diversity based on their higher degree of ploidy. Our findings suggest that formation of multiple DSBs allow complexed genome rearrangements and phenotypic diversification, particularly in combination with WGD.

## Results

### Induction of genome rearrangements by multiple DSB formation.
To monitor the process of genome rearrangement, we newly developed a method involving TaqI-aided genome rearrangement (hereafter referred to as the "TAQing system"), as illustrated in Fig. 1a. We first verified the TAQing system in vegetative haploid S. cerevisiae cells (YPH499). TaqI is a four-base endonuclease that specifically recognizes the TCGA palindrome and can theoretically cleave genomic DNA once every 256 base pairs. This TCGA sequence can help us consider the causal relationship between rearrangement events and the adjacent potential DSB sites, which are difficult to estimate in radiation-based or mutagen-based conventional mutagenesis. TaqI activity is temperature-dependent and only functions at temperatures >37 °C (Supplementary Fig. 1a). The full-length TaqI gene was conditionally expressed in the presence of $Cu^{2+}$ from vectors harboring a $Cu^{2+}$-inducible promoter (Fig. 1b) and activated one-time by transient heat treatment. Heat treatment tentatively induced random formation of DSBs at multiple chromosomal sites, as confirmed by Rad51 foci formation (Supplementary Fig. 1b), a DSB-dependent band shift in the DNA damage checkpoint proteins Rad53 and Rad9 (Fig. 1c), and degradation of chromosomal bands in pulse-field gel electrophoresis (PFGE) (Fig. 1d). Based on the degree of broken DNA fragments according to PFGE analysis, we estimated that the DSB frequency was comparable to that in diploid meiotic yeast cells[16], which reportedly produce ~200 DSBs per cell.

This condition of TAQing with heat-activation resulted in a reduced number of substantially viable cells (~20% viability in haploids and 40% in diploids; Fig. 1e,f). Introduction of an inactive TaqI variant (dTaqI; with a D142A mutation) into yeast cells resulted in no such effects on viability (Fig. 1e), indicating that the reduced viability was caused by TaqI DNA cleavage.

To determine whether activated TaqI could induce homologous recombination (HR) in diploid cells, we monitored interallelic gene conversion of the arg4-nsp/arg4-bgl heteroalleles, finding a marked (18-fold) increase in HR frequency (Fig. 1g). In addition, loss-of-heterozygosity assays (Fig. 1h) showed that TaqI heat activation markedly induced gene conversion or unequal crossing over.

PFGE experiments revealed that 28.1% (9/32) of activated TaqI-treated ("TAQed") diploids exhibited changes in chromosome size, whereas this was observed in only 3.1% (1/32) of mock controls. Importantly, no visible rearrangements were observed in TAQed haploid clones (0%, 0/16; Supplementary Fig. 2a and Table 1), suggesting that yeast diploids underwent genome rearrangements much more efficiently than haploids. These results indicated that the TAQing system was capable of inducing large-scale genome rearrangements more efficiently in yeast diploids than in haploids. A possible explanation for this is that haploids are less tolerant of gross genome rearrangements than diploids because of the lower redundancy of their genetic information.

### Rearrangements between homologous sequences and DSB ends.
Of the 2307 colonies of TAQed cells, we selected 178 mutants which formed smaller colonies (<20% of the area). In total, 131 of these mutants showed altered cell morphology revealed by microscopic observation (1.56 ± 0.43-fold larger cell area, mean ± SD; Supplementary Fig. 2c,d). Additionally, 11/13 mutant isolates exhibited altered chromosomal size detectable by PFGE analysis (Fig. 1i), suggesting that the massive genome rearrangements frequently accompanied phenotypic changes. To identify the precise genomic alterations, 13 mutant isolates exhibiting altered morphology were further examined by combined long-read/short-read high-throughput sequencing and tiling-array analyses (Supplementary Fig. 2c).

The fused parental diploid WT14 (MATa/a) strain shared 0.7% single nucleotide variations (SNVs)[17], which enabled the identification of chromosomal-rearrangement sites in the mutants. Accordingly, the genome sequences of the parental strain were reassembled from the high-throughput resequencing

data generated by short-read Illumina MiSeq (>500 coverage; Illumina, San Diego, CA, USA) and long-read PacBio RSII (>87 coverage; Pacific Biosciences, Menlo Park, CA, USA) sequencing.

Comprehensive resequencing of the 13 TAQed mutant isolates (MiSeq: 73–254 coverage) showed an average of 1.3 SNVs/strain

(Fig. 2a and Table 1; Supplementary Table 1), whereas SNVs were absent in the control diploid without TaqI activation. It should be noted that the frequency of TaqI-mediated canavanine-resistant colony formation, which supposedly reflects mutation frequency, was substantially reduced (~50%) in *rev3Δ* (error-prone DNA

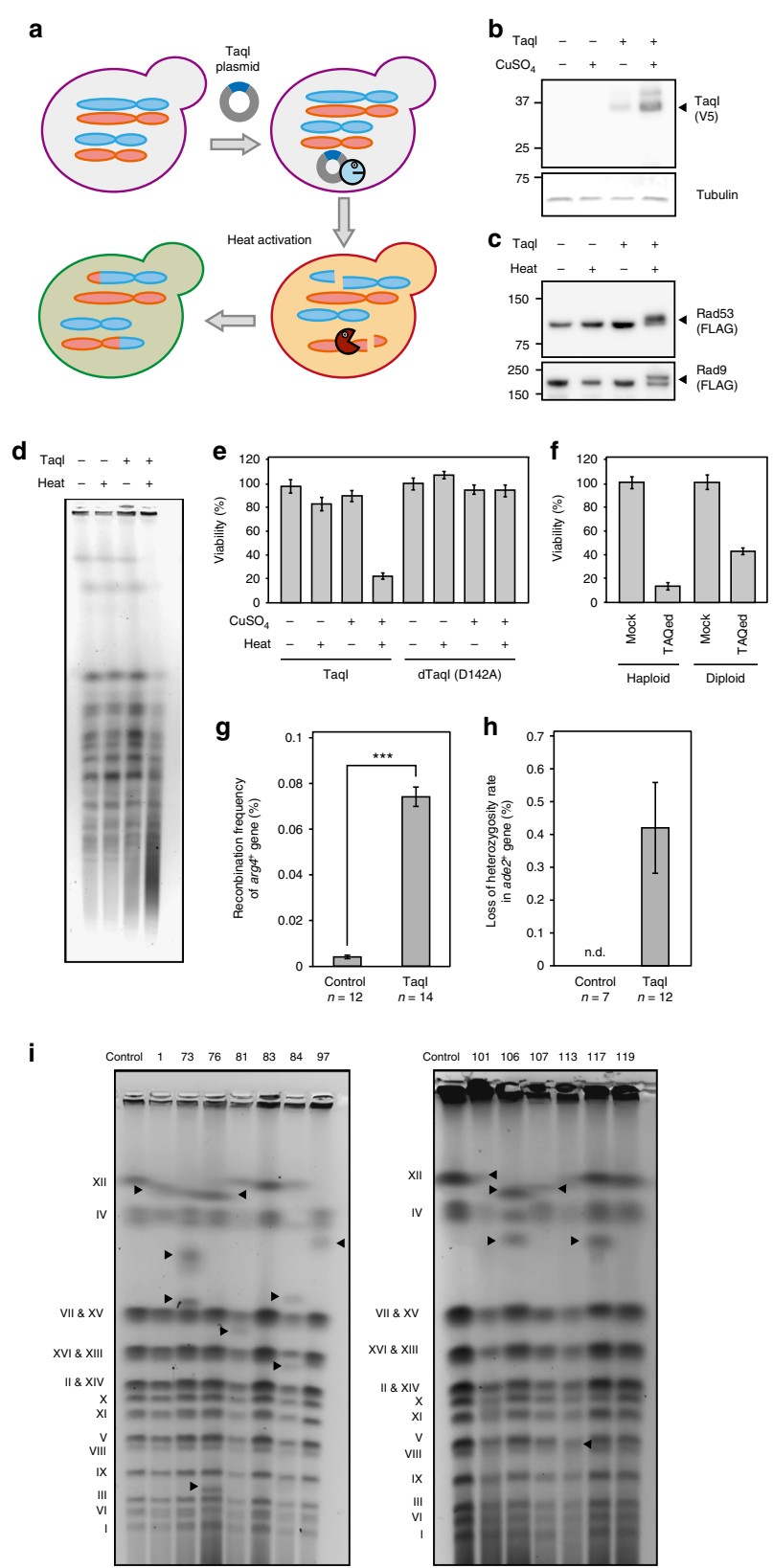

polymerase) mutant cells[18] (Fig. 2b). This suggested that these mutations, as reported previously[19], were at least partly induced by error-prone DNA repair of TaqI-mediated DSBs that proceeded not via canonical non-homologous end-joining (C-NHEJ), but via HR or microhomology-mediated end-joining (MMEJ; or alternative NHEJ, A-NHEJ) associated with a rather long resection.

We found that TaqI activation promoted various types of chromosomal aberrations (Fig. 2a,c,d; Supplementary Figs. 3 and 4, and Table 1), none of which were detected in the mock control. We detected 40 non-reciprocal inter-homolog short gene conversions (SGCs; 0.7–30 kb; average length: 7.0 kb) (Fig. 2a and Table 1), confirming that yeast cells underwent HR-based repair very efficiently. Notably, all six non-homologous translocation (TL) breakpoints were at TaqI-recognition sequences (Fig. 2d, e; Supplementary Table 1) without any deletions, suggesting that the cohesive two-nucleotide overhangs at TaqI-generated DSBs could be directly joined in vivo, possibly without any resections of DSB ends. Such a high incidence of unequal TLs after DSB formation in mitosis is distinct from their scarce occurrence after natural Spo11-induced meiotic DSB formation in S. cerevisiae reversion assays[20].

Two isolates harbored chromosomes with fused TL breakpoints within Ty-retrotransposable elements, and another isolate had a chromosome fusion within a repetitive ribosomal DNA (rDNA) region (Fig. 2c and Table 1). This was consistent with previous studies reporting that chromosomal TLs in Saccharomyces-like species are often associated with Ty elements or repetitive sequences[21]. These results indicated that the TAQed yeast cells were susceptible to massive chromosomal rearrangements, which occurred between homologous sequences or unprocessed DSB ends.

We also examined the three-dimensional (3D) positioning of loci surrounding the TL breakpoints by referring to the chromosomal contact information revealed by a previous Hi-C study[22]. According to Hi-C data, the average distance of any random two-chromosome loci in S. cerevisiae was calculated at 0.86 μm. The distances for the four pairs of six chromosomal loci for C-NHEJ-joined unequal TLs were smaller than this average, whereas those for the remaining two pairs were near this average (Fig. 2f,g). The average distance of all six C-NHEJ-joined unequal translocated loci pairs was 0.57 μm. Notably, the pairs of chromosomal-breakpoint loci for TL-Chr4:13, TL-Chr5:15, and TL-S799Chr6:YPH499Chr6 were located particularly close to each other (Fig. 2f,g). These results suggested that C-NHEJ-mediated TLs occurred between loci in relatively close proximity. It should be also noted that two pairs of breakpoint loci for TL-Chr10:15 and TL-Chr4:16, which were assumed to result from HR events between Ty-repetitive sequences, were located at relatively distant chromosomal domains (Fig. 2f,g).

**TAQed yeast genomes confer marked phenotypic diversity**. In addition to variability in the cell size and morphology (see Fig. 3a; Supplementary Fig. 2d), we also detected substantial non-morphological qualitative changes in the TAQed yeast diploids. For example, 6.2% (11/178) isolates exhibited hyper-flocculation

or hypo-flocculation phenotypes (Fig. 3b,c; Supplementary Fig. 2c). This TAQing-induced phenotypic diversity led us to further examine whether different traits derived from two strains could be stably combined, as the two strains are subjected to cell fusion, followed by TAQing treatment.

To investigate this, we fused a xylose-utilizable haploid strain (capable of growing on xylose-containing medium) and a thermotolerant haploid strain (which can grow at 40 °C) to obtain a non-TAQed diploid cell-fusion strain. This fused non-TAQed strain initially exhibited both xylose-fermentation ability and thermotolerance; however, after multiple passages, one of such combined phenotypes was lost, as it ceased to grow on xylose-containing medium at 40 °C.

We then heat treated the freshly prepared fused non-TAQed diploid to induce TAQing, followed by single-colony isolation on xylose-containing medium at 40 °C. As a control, non-TAQed fused strains were subjected to the same procedure in the absence of TAQing (Supplementary Fig. 5a).

Notably, only the TAQed diploid continued to grow stably under the combined stress environment during selection (Fig. 3d), suggesting that the non-TAQed fusion lost one of the combined traits during preculture. It should be noted that the ethanol-production efficiency of the TAQed mutant in xylose-containing medium was as effective as that in glucose-containing medium (Fig. 3e). We subsequently confirmed that such combined phenotypes could be stably inherited over 10 times of passages in the TAQed fusion strain, whereas the non-TAQed fused diploid easily lost one of the favorable phenotypes (Supplementary Fig. 5b–d). These results showed that genome rearrangements induced by the TAQing system substantially contributed to phenotypic diversification and stable inheritance of useful combined quantitative traits after at least 10 passages.

**TAQing in plants**. To monitor the rearrangement process after multiple DSB formations in multicellular organisms, we tested the TAQing system using diploid A. thaliana plants harboring the TaqI-expression vector along with the constitutive CaMV 35S promoter [TaqI+(2n)]. To examine the effects of WGD, diploid

**Table 1 Summary of mutation types in Fig. 2a**

| Type of rearrangement | Numbers of events (in 13 strains) | Note |
|---|---|---|
| SNVs | 17 | |
| Insertion | 6 | |
| Deletion | 1 | |
| Aneuploid | 7 | 4 losses |
| Break-induced repair | 10 | 2 in rDNA regions |
| Short gene conversion | 40 | |
| Translocation (non-homologous) | 6 | |
| Translocation (homologous) | 3 | 2 in Ty-like regions 1 in rDNA region |

**Fig. 1** DSB formation by heat-activated TaqI. **a** Schematic diagram of the TAQing system. A TaqI-expression vector was introduced into yeast cells, followed by transient heat activation of TaqI (inactive, blue circle; active, red circle) to generate DSBs in vivo. Red and blue ovals represent homologous chromosomes. **b** Cu²⁺-induced TaqI-expression analysis in haploid (YPH499) cells. An arrowhead indicates TaqI protein. **c** Immunodetection of DSB-induced Rad53 and Rad9 band shifts. Arrowheads indicate the phosphorylated forms of Rad53 and Rad9. **d** Chromosome-break analysis by pulse-field gel electrophoresis (PFGE). **e** Cell viability after TaqI activation in haploid cells (n = 5). dTaqI; dead (inactive) TaqI mutant. **f** Cell viability after TAQing in haploid and diploid cells (n = 5). Mock; no CuSO₄ and no heat activation. **g** Homologous recombination rate at arg4-nsp/arg4-bgl heteroalleles in diploid (MJL1720) cells. **h** Loss-of-heterozygosity frequency of the ADE2 marker in fused diploid (WT14) cells. Data represent the mean ± SEM. ***P < 0.001; Welch's t-test. **i** PFGE analysis of chromosome sizes in TAQed diploid strains exhibiting altered cell morphologies. Arrowheads indicate chromosomes with size difference. Chromosome number is shown on the left

plants were treated with colchicine to produce tetraploids, followed by transformation of the TaqI-expression vector [TaqI$^+$(4n)](Supplementary Fig. 6). These plants were incubated at 37 °C for 24 h to activate TaqI (Fig. 4a). We observed an ~10-fold upregulation in expression of the DSB repair gene *AtRAD51*[23]

was upregulated ~10-fold in TAQed diploid [TaqI$^+$(2n)] and tetraploid [TaqI$^+$(4n)] plants (Fig. 4b). A β-glucuronidase (GUS) recombination reporter assay[24] revealed a marked increase (>20-fold) in HR events in cotyledons of the TAQed plants (Fig. 4c). While both diploid and tetraploid plants exhibited growth

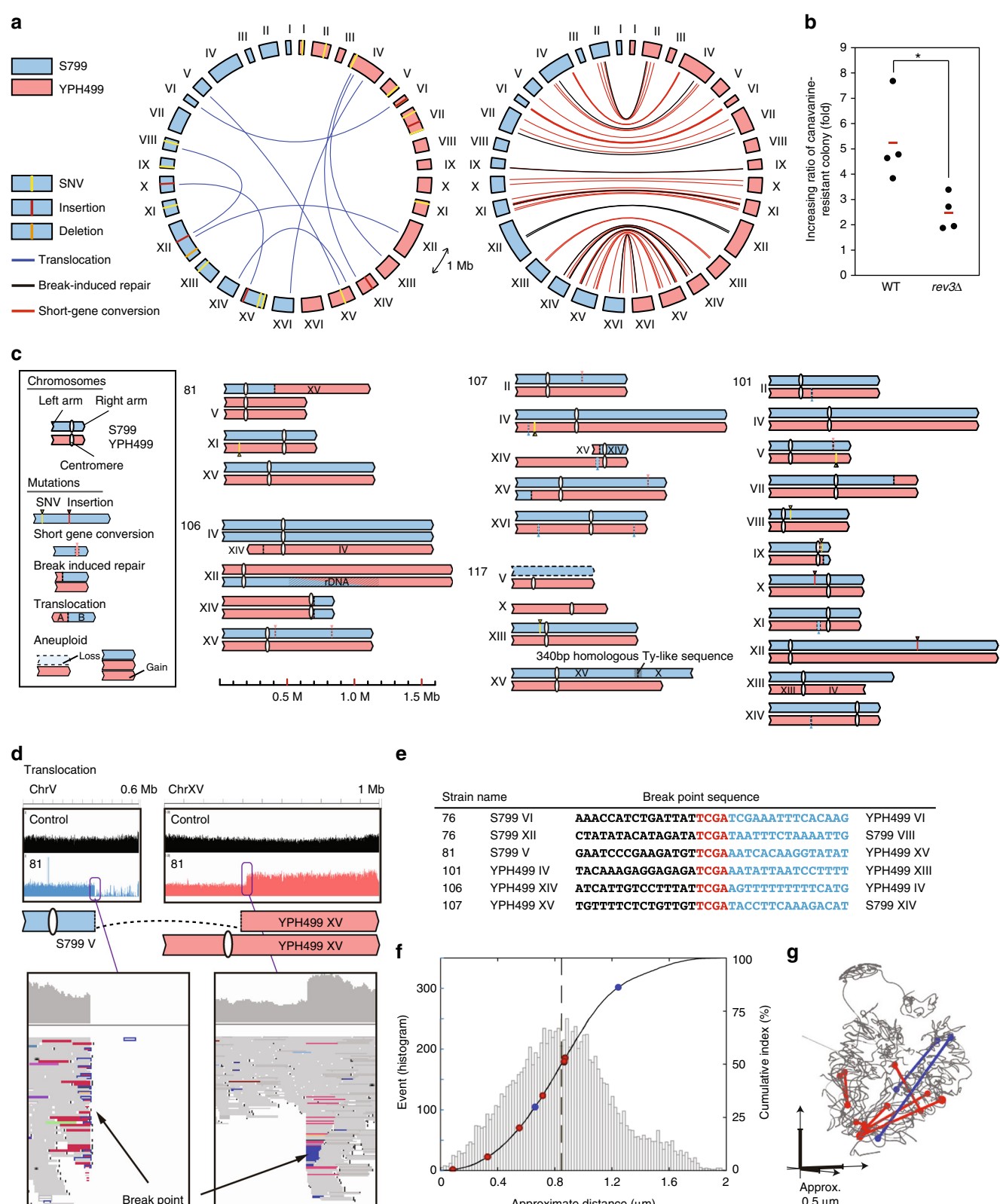

retardation, likely due to DNA damage-induced cell arrest[25], most plants were fertile and produced viable seeds with slightly lower yields (Supplementary Fig. 7).

Some progenies of TAQed plants showed changes in both morphology and biomass (Fig. 4d; Supplementary Fig. 8a). The stem lengths of TAQed mutants #13 and #19 were 30% higher (Fig. 4d) than that observed in the mock control. Additionally, compared to wild-type plants, the TAQed mutants showed a 40% increase in biomass (dry weight of aerial parts) (Fig. 4e)[26]. Importantly, the relative distribution of stem lengths in the 81 TAQed tetraploid (TQ4) plants was significantly broader ($P <$ 0.01, $F$-test) than that in the 40 TAQed diploid (TQ2) plants (Supplementary Fig. 8b), with the variance in the stem-length distribution in the TQ4 plants ($\sigma^2 = 6.65 \times 10^{-2}$) is 4.3-fold greater than that in TQ2 plants ($\sigma^2 = 1.53 \times 10^{-2}$). Thus, TAQing-induced genome rearrangement leads to greater phenotypic diversification in tetraploids than diploids.

Notably, plants undergoing massive chromosomal rearrangements exhibited changes in leaf size (Fig. 5a) that were successfully transmitted to offspring (Fig. 5b). Specifically, in the case of TQ4_e progenies, the fusion of Chr1 and Chr4 might affect the plant size. These data indicated that the new quantitative traits acquired by the TAQing system were stably inherited and retained over at least three generations.

**More highly complex rearrangements occur in tetraploids**. We then conducted combined tiling-array analysis and short-read high-throughput resequencing of TAQed plants, detecting ~25 mutations/genome in TQ2 and TQ4 plants (Fig. 6a, Supplementary Data 1), which was 2–4-fold higher than that observed in wild-type plants and mock controls[27]. We also observed increases in the number of insertions/deletions (InDels) in TAQed progenies (Fig. 6a).

On the other hand, 11.3% (9/80) of TQ4 plants showed 22 large CNV events (CNV rate per individual: 0.28) (Fig. 6b; Supplementary Fig. 9a; and Table 2); however, no CNVs (0/16) were found in the mock-control tetraploid plants, indicating that TAQing-mediated DSBs induced the observed higher rate of genome rearrangement in tetraploids. Importantly, large CNV-associated chromosomal rearrangements were not observed in the TQ2 plants (0%, 0/12; Supplementary Fig. 9b; Table 2).

Moreover, 46.3% (37/80) of TQ4 plants showed 42 aneuploidies (chromosomal CNVs; aneuploidy rate per individual, 0.53), whereas some chromosomal CNVs (3/16) were detected in mock-control tetraploid plants (aneuploidy rate per individual, 0.19). Similar to previous results, no chromosomal CNVs were detected in 12 TQ2 plants (aneuploidy rate per individual: 0.0) (Table 2). These results suggested that TAQing-induced aneuploidies and CNVs much more effectively in tetraploids than in diploids.

**TLs are often found in the vicinity of transposable elements**. Chromosome-rearrangement sites were identified by sequencing the genomes of four TQ4 plants harboring large CNVs (Fig. 6c; Supplementary Fig. 10 and Table 2). We detected eight interchromosomal TLs, one intrachromosomal TL, three large (0.27, 0.49, and 2.03 Mb) deletions, and one tandem duplication. Additionally, large-scale chromosomal rearrangements were frequently found in TaqI-recognition sequences (Fig. 6d), suggesting that TaqI-induced DSBs caused these genome rearrangements via cohesive two-nucleotide overhangs, as observed in *S. cerevisiae* (Fig. 2f). Moreover, 30.7% (4/13) of large CNV events contained microhomology regions (2–3 nucleotides) lacking the TaqI-cleavage site, indicative of error-prone MMEJ/A-NHEJ events (Fig. 6d). In the other cases, similarity between TL-breakpoint sequences was not detected (Fig. 6d). Furthermore, 15.4% (4/26) of TL breakpoints were detected within transposable elements, such as those of the RC/Helitron and LTR/Gypsy families (Supplementary Table 2). A previous study suggested that these transposable elements drive plants evolution by genome shuffling[28,29].

## Discussion

Our results demonstrated that one-time activation of genome-wide multiple DSB formation is sufficient to efficiently induce large-scale genome rearrangements and phenotypic diversification. These findings suggested that simultaneous formation of multiple DSBs can promotes gross genome rearrangement. We observed that the degree of rearrangements was much more pronounced in cells with higher ploidy. In yeast, massive genome rearrangement was observed in diploids, whereas little chromosomal aberration was detected in haploids. In *Arabidopsis*, large-scale CNVs and aneuploidies were not detected in 16 diploid TAQed plants, whereas tetraploid TAQed plants more frequently induced complex CNV-associated genome rearrangements (42 CNVs in 37 TQ4 plants; Fig. 6b; Table 2; Supplementary Fig. 9a). These data suggested that increased in ploidy allowed higher genomic plasticity[30] and promoted more complex genome rearrangements. Under the activation conditions used in this study, the TAQing system was assumed to generate nearly 200 simultaneous DSBs, which might not be optimal for the investigation of the DSB effects in cells with lower degrees of ploidy. Usage of different restriction enzymes or activation conditions for TAQing might address this problem.

We observed that WGD-enhanced phenotypic diversity in TAQed yeast and plant cells. For example, the stem-length divergence was 4.3-fold larger in tetraploid TAQed plants than in diploid TAQed plants (Supplementary Fig. 8b). One possible reason for this is that the high redundancy in genetic information acquired following WGD allowed more complexed genome reorganization, whereas such rearrangements tend to cause lethal effects in cells with limited redundancy of genetic information. Therefore, haploids with rearrangements might be quickly eliminated. Another explanation is that WGD alone can promote phenotypic diversification. It is possible that combinatorial patterns of CNVs can be more divergent in cells with higher ploidy.

The increased allowance of phenotype heterogeneity following WGD is consistent with previous studies on natural-selection processes in selfing species. Inbred yeast and selfing plant genomes likely emerge from hybrid fusions or WGD in combination with

**Fig. 2** Chromosome structures in TAQed yeast strains. Genome variations in the 13 TAQed strains derived from a heterozygous diploid strain (S799 and YPH499 parental strains). **a** Circular diagram of SNVs, InDels, short-gene conversions (SGCs, red), break-induced repairs (BIRs, black), and translocation (TLs, blue) in the 13 TAQed strains. YPH499 chromosomes, red arcs; S799, blue arcs. **b** Increasing ratio of canavanine-resistant colonies following TAQing treatment. The red line indicates mean of four independent experiments. *$P < 0.05$; Welch's $t$-test. **c** Schematic diagrams of rearranged chromosomes in aneuploid TAQed strains. **d** Chromosome-wide mapping examples for control (black) and TAQed strains (S799, blue; YPH499, red) with a typical chromosomal rearrangement, the TL. Magnified views around the TL breakpoint is shown. **e** TL-breakpoint sequences in five TAQed strains. **f** Distance distribution (histogram and cumurative index curve) of randomly selected two alleles or loci pairs for TL events in the previous Hi-C data. Red and blue dots represent loci pairs for C-NHEJ-mediated and HR-mediated TLs, respectively. **g** 3D location of loci pairs associated with TL events. Red and blue dots linked with lines represent loci pairs for C-NHEJ-mediated and HR-mediated TLs, respectively

genomic rearrangements. For example, pentaploid or hexaploid brewer's yeast (*Saccharomyces pastrianus*) was likely formed from the hybridization of *S. cerevisiae* and the wine yeast *Saccharomyces eubayanus*[31–33]. Moreover, polyploidization and segmented chromosomal duplication in *S. cerevisiae* cause phenotypic changes[34–36].

Therefore, we speculated that the phenotypic diversification of *A. thaliana*[37–39] and *Brassica rapa*[11,12] acquired during natural selection might have involved similar mechanisms.

It is intriguing that some TLs occur via C-NHEJ of DSB ends or HR between repetitive sequences. TaqI generates DSB ends

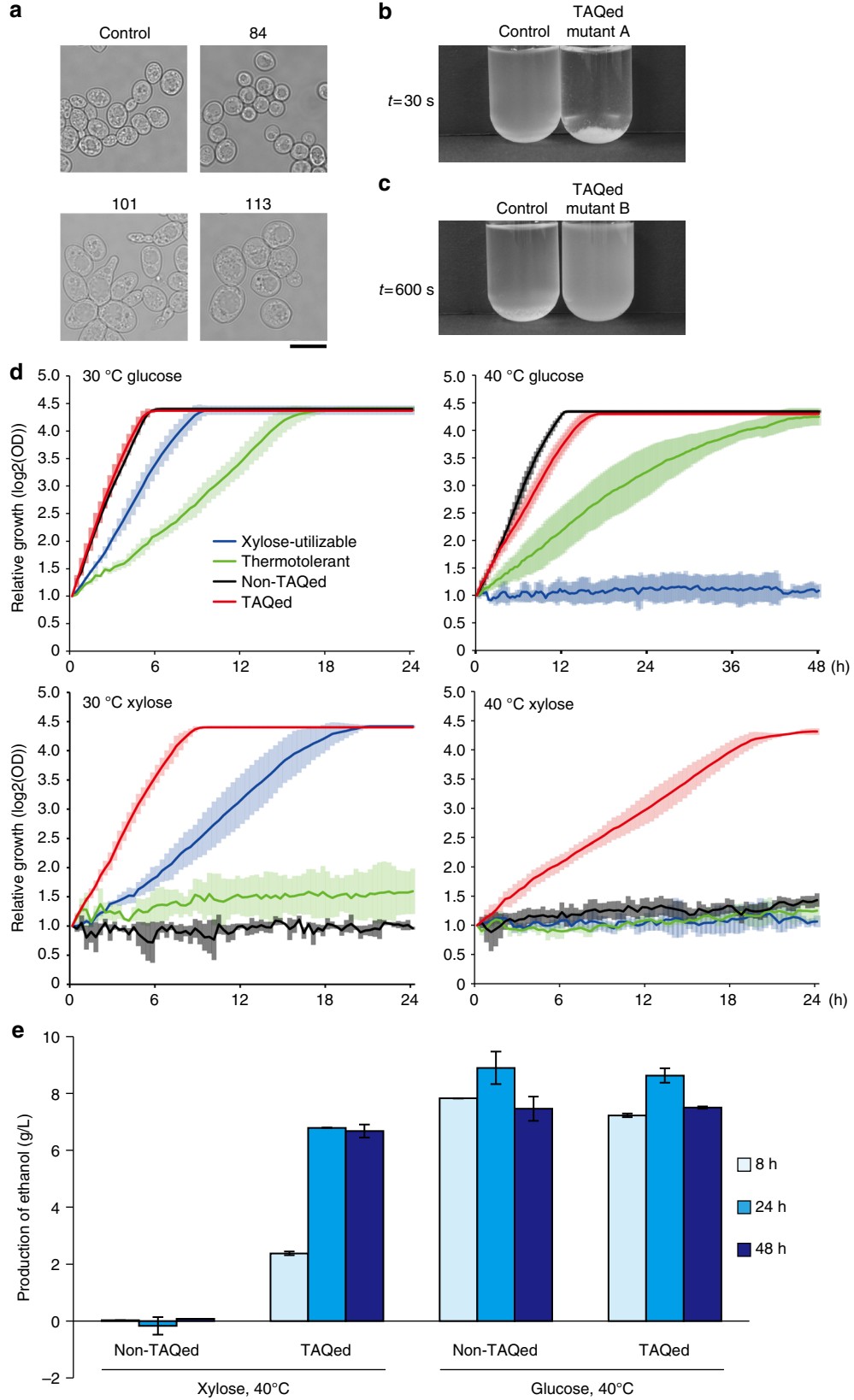

with two-bases cohesive 5′ overhangs. Our results indicated that in living yeast and plant cells, many of such DSB ends are directly joined independent of HR without any resection of the DSB ends. We speculated that the majority of TaqI-induced DSBs were promptly joined without causing TLs or any sequence changes; however, these events cannot be detected by DNA sequencing. This idea seems consistent with a recent observation that HR in yeast rDNA regions is dispensable for DSB repair at arrested forks, and that suppression of end resection is important for protecting DSBs from genome rearrangements[40]. It would be instructive to investigate whether the CRISPR-associated DNA-cleaving enzyme Cpf1[41], which generates staggered DNA ends, could induce TLs between chromosomal sites of interest. Some TL breakpoints are found in homologous and/or repetitive sequences, such as retrotransposons, transposons, and rDNA. These types of rearrangements are postulated to induce large-scale genomic evolution. This idea is supported by a previous observation that rearrangements among chromosomes of *S. cerevisiae*-related species are often associated with retrotransposons[42]. This tendency could be extended in more complex multicellular organisms with an abundance of repetitive sequences. In the somatic cells of multicellular organisms, repetitive sequences or microhomology is often found at breakpoints associated with deleterious chromosomal rearrangements, some of which promote cancer progression.

The TAQing system offers several practical benefits. For example, the breeding of fermentative microorganisms sometimes requires the generation of hybrids from strains with different useful traits. While such fused strains can easily lose one of the useful traits during multiple rounds of passage, rearrangements induced by the TAQing system are highly stable and enable long-range retention of multiple beneficial traits.

The reason for the stable inheritance of acquired traits in TAQed strains remains unknown. In strains obtained by simple cell fusion, chromosomes harboring genes "favorable" for the acquired traits might often be eliminated during multiple rounds of cell divisions, thereby causing trait instability. On the other hand, TAQed stains carry recombined chromosomes with increased copies of favorable genes as a consequence of CNVs, which ultimately reduces the risk of trait loss over multiple passages.

The TAQing system also allows researchers to breed yeast strains for industrial use, even sterile parental strains (e.g., brewery yeast and *Candida utilis*). Additionally, the TAQing system can overcome the bottlenecks of conventional breeding methods, such as mutagen treatments, which results in relatively limited genetic alterations (e.g., point mutations), and radiation-based mutagenesis, which has limitation of genomic rearrangements due to the lethal effects of irradiation.

Another important approach to artificially evolving the yeast genome is "SCRaMbLE" (synthetic chromosome rearrangement and modification by LoxPsym-mediated evolution)[43]. This method is based on the synthetic yeast genome-engineering project called Sc2.0, in which large segments of the *S. cerevisiae* genome are systematically replaced with synthetic DNA. In SCRaMbLE, LoxPsym sites are inserted into the 3′ untranslated region of each

non-essential gene to permit various chromosomal rearrangements, such as translocations, inversions, and deletions, at each site. The SCRaMbLE-induced complex rearrangements enable artificial evolution and synthesis of the yeast genome at will.

Compared with SCRaMbLE, the TAQing system is not good at systematically altering the genome or specifically altering it between coding segments. However, this system enables the shuffling of TaqI-cleavage sites in DNA elements across multiple organisms in the absence of a requirement to artificially synthesize chromosomes with LoxPsym sites. In addition, the use of CRISPR–Cas9 or different restriction endonucleases that recognize other DNA sequences allows alteration of sites of interest and the extent of subsequent genome rearrangements. Both SCRaMbLE and TAQing represent powerful and complementary tools for artificial genome evolution.

In conclusion, our findings confirmed that genome rearrangements can be promoted after WGD to markedly enhance phenotypic diversification. These finding also raises new questions to be addressed in future studies, including the degree of genome rearrangements allowed after multiple rounds of TAQing, and the natural source of DSBs in adaptive evolution. Additionally, TAQing can be used as a new tool to study genome instability and genetic networks or as a methodology for metabolic engineering to generate new genomic sequences that improve pathway flux in many organisms.

## Methods

**Yeast strains and culture.** Yeast strains YPH499 (S288c-derived haploid; *MATa ura3-52 lys2-801 ade2-101 trpl-Δ63 his3-Δ200 leu2-Δ1*) and S799 (SK1-derived haploid; *MATa ura3 lys2 ho::LYS2 leu2Δ arg4-bgl cyh2-z*) were used as wild-type strains. Cell-fusion diploid strains (WT14; TAQed mutants 1, 73, 76, 81, 83, 84, 97, 101, 106, 107, 113, 117, and 119) were generated by cell fusion of YPH499 and S799, followed by TaqI activation. All yeast strains are also listed in Supplementary Data 2. Yeast cells were cultured in yeast extract–peptone–dextrose (YPD), synthetic defined (SD), SD/monosodium glutamic acid (MSG), or minimal (MM) medium at 30 °C. The composition of MM was 20 g glucose, 2 g $KH_2PO_4$, 1 g $MgSO_4$, 1 g ammonium sulfate, and vitamins (0.002 mg biotin, 0.4 mg calcium pantothenate, 0.002 mg, folic acid, 0.4 mg niacin, 0.2 mg p-aminobenzonic acid, 0.4 mg pyridoxine HCl, 0.2 mg riboflavin, 0.4 mg thiamine HCl and 2 mg inositol) in 1 L.

**Yeast strain construction.** YPH499 and derivatives were used in haploid-cell experiments. Strains were transformed with an hphMX4-selectable vector harboring *RAD5* or *RAD9* with C-terminal 6×His and 3×FLAG tags, respectively. The primers NT208 (5′-GGGCAAAATTGGACCAAACCTCAAAAGGCCCCGA-GAATTTGCAATTTTCGTCCCACCACCATCATCATCAC-3′) and NT209 (5′-TCTGAGTATTGGTATCTACCATCTTCTCTCTTAAAAAGGGGCAGCATTT-TACTATAGGGAGACCGGCAGATC-3′) were used for *RAD53*, and NT206 (5′-ACGATGATATTACGGGACAATGATATATACAACACTATTTCTGAGGTTA-GATCCCACCACCATCATCATCAC-3′) and NT207 (5′-AGAATCTCTAAAT TTTTTTTTATTTAATCGTCCCTTTCTATCAATTATGAACTATAGGGA-GACCGGCAGATC-3′) were used for *RAD9*. DNA fragments were introduced into YPH499 cells, which were then selected for based on hygromycin B resistance. Correct integrations were verified by polymerase chain reaction (PCR). *REV3* mutants were generated similarly with the primers NT132 (5′-TGTGCCCAAGTTCA-TAAAACGCTGGAAGTAAAAATTAGGGCATCCTTTAAGGGTTAATTAAG GCGCGCCA-3′) and NT133 (5′-ATTACAACATGTTGCA-GAAGTTCCTACTCTTGTTATGTTTTCCACTTTTGCATCGATGAATTC-GAGCTCG-3′). YPH499 and S799 were fused to prepare WT14, as previously described[44]. Protoplasting solution was modified to MP buffer containing 28 U/mL

---

**Fig. 3** Phenotypic diversification in TAQed yeast strains. **a** Phase contrast images of representative TAQed mutants. Scale, 10 μm. **b,c** Heterogeneity in flocculation phenotype in the TAQed strains (a fusion of YPH499 and S799). Images of hyper-flocculation **b** and hypo-flocculation **c** mutants at the indicated times after mixing. Control, non-TAQed strain. **d** Growth curves of xylose-utilizable haploid (W700M2), thermotolerant haploid (N44D), non-TAQed, and TAQed fused diploids (SNT17 and SNT11-2×40, respectively) in glucose-containing or xylose-containing media at 30 or 40 °C. Optical density at 600 nm ($OD_{600}$) of each independent culture initiating from a low cell density ($<OD = 0.001$) was measured at every minute. We set the $t = 0$ point as $OD_{600} = 0.2$ when cells undergo exponential growth. We plotted averages of each 10 min interval of three biological replicates along with standard deviations (vertical bars). Non-TAQed fused diploid cells grew faster with glucose at 40 °C than other strains, but did not with xylose at 40 °C. Only TAQed strain proliferates well in the presence of xylose at 40 °C. **e** Ethanol production in culture supernatant was measured after culturing in glucose-containing or xylose-containing media at 40 °C for 8, 24, and 48 h. The average of three individual replicates, error bars are ± SD

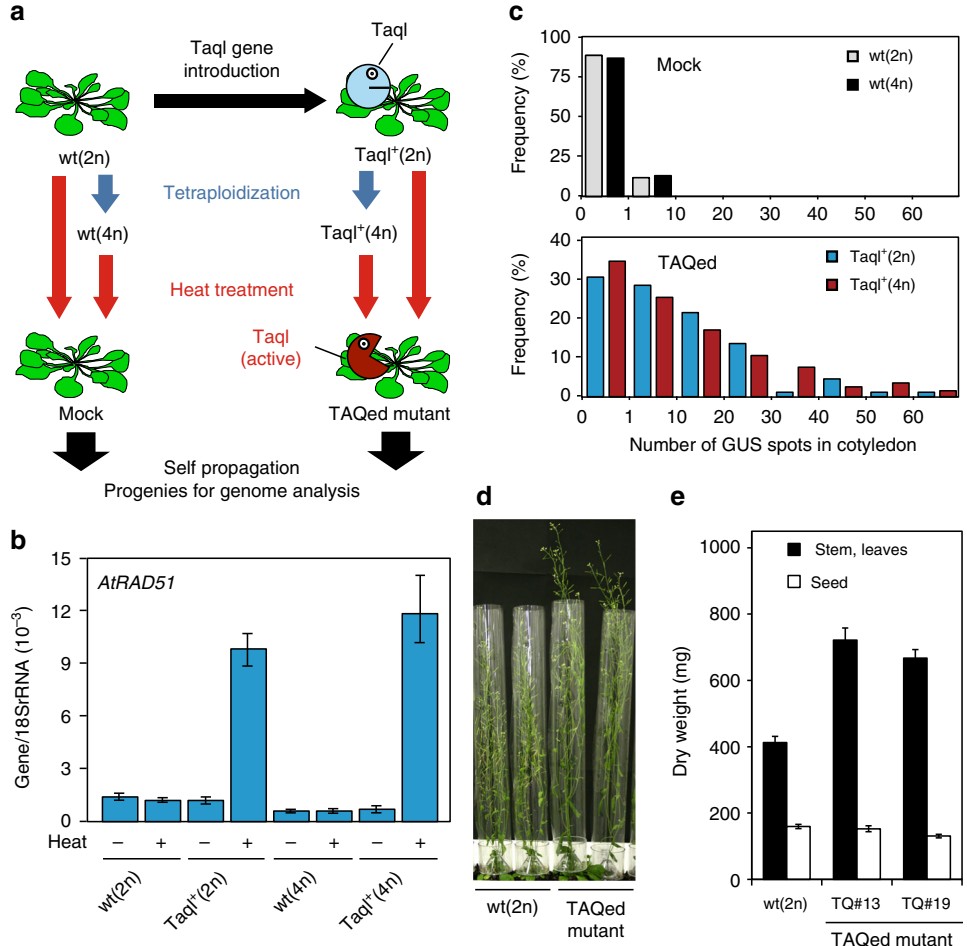

**Fig. 4** TaqI-induced DSBs and mutations induced in *Arabidopsis* plants. **a** Experimental scheme of induced genome rearrangements in diploid and tetraploid plants. Diploids and tetraploids harboring TaqI transgene are indicated as TaqI+(2n) and TaqI+(4n), respectively. **b** *AtRAD51* mRNA expression following TaqI activation. Data represent the mean ± SEM ($n = 3$). **c** Frequency distribution of blue spots by GUS-reporter gene recombination in cotyledons of TAQed diploid [TaqI+(2n)] and tetraploid [TaqI+(4n)] plants. **d,e** The biomass growth phenotype increased in TAQed mutants. **d** Image shows control and TAQed mutant (#13, left; #19 right) plants grown in soil for 54 days, **e** dry weight of plants grown in soil for 14 weeks. Data represent the mean ± SEM ($n = 20$)

zymolyase (Seikagakukogyo, Kogyo Co. Ltd., Tokyo, Japan), 25 μL/mL glusulase (Perkin Elmer, Waltham, MA, USA), and 1.7% β-mercaptoethanol. This fused diploid was the parental strain of TAQed mutant strains 1, 73, 76, 81, 83, 84, 97, 101, 106, 107, 113, 117, and 119. MJL1720 (*MATa/α ura3/ura3 lys2/lys2 ho::LYS2/ho::LYS2 leu2Δ/leu2Δ arg4-bgl/arg4-nsp cyh2-z/cyh2-z*)[45] and was used in HR assays.

**Xylose-utilizable and thermotolerant yeast preparation**. Expression plasmids were constructed to overexpress the following genes of the nonoxidative pentose–phosphate pathway: *XKS1*-encoding xylulokinase, *RKI1*-encoding ribulose 5-phosphate isomerase, *RPE1*-encoding ribulose 5-phosphate epimerase, *TKL1*-encoding transketolase, and *TAL1*-encoding transaldolase. The integration vector pXhisHph-HOR7p-ScXK which is targeted to the *HIS3* locus in chromosome XV, was used for the construction of the *XKS1* overexpression plasmid[45]. The integration vector pXAd3H-HOR7p-ScTAL1-ScTKL1, which is targeted to the upstream region of *ADH3* in chromosome XIII, was used for the construction of *TAL1*- and *TKL1*- overexpression plasmids[45]. The integration vector pXGr3L-HOR7p-ScRPE1-ScRKI1, which is targeted to the *GRE3* locus in chromosome VIII, was used for the construction of the *RPE1*-overexpression and *RKI1*-overexpression plasmids[46]. All the genes were expressed under the control of the *HOR7* promoter and the *CYC1* terminator.

A pRS524-HOR7p-PiXI vector that enabling multicopy chromosomal integration of the *Piromyces* sp. E2-derived XI gene (*PiXI*) onto a chromosome was constructed, with the vector containing the *PiXI* gene (GenBank: AJ249909) along with the *HOR7* promoter added 5′ of the gene and a *TDH3* terminator added to the 3′ region. Furthermore, to promote HR into yeast genome, R45 and R67 sequences homologous to the *rRNA* gene (rDNA), and as a marker, the gene sequence of a TRP1d marker and exhibiting low levels of expression through deletion of a portion of its promoter.

These R45 and R67 sequences enable multicopy integration of the sequence containing *PiXI* to the rDNA locus on chromosome XII[47].

A W303-1B-based strain overexpressing pentose–phosphate pathway genes was constructed as follows. First, the *ADE2* was amplified by PCR from the genomic DNA purified from *S. cerevisiae* S288C using the primer pair ADE2+1F (5′-ATGGATTCTAGAACAGTTGGTATATTAGG-3′) and ADE2+1716R (5′-TTACTTGTTTTCTAGATAAGCTTCGTAACC-3′). The amplification product was used to transform the W303-1B strain that complemented the *ade2* mutation. The resulting strain (W303-1BA) was transformed with linearized DNA fragments obtained from the *Sse8387*I-digested plasmids pXhisHph-HOR7p-ScXK, pXAd3H-HOR7p-ScTAL1-ScTKL1, and pXGr3L-HOR7p-ScRPE1-ScRKI1. The resulting strain was named W600[48]. Next, a fragment was obtained by digesting ~1 μg of the pRS524-HOR7p-PiXI with a restriction enzyme *Fse*I, followed by transformation of W600 with the linearized the DNA fragment to create the xylose-utilizable strain was named W700M2[48]. N44D, an *S. cerevisiae* NBRC1444-derived haploid strain (*MATα*), was used as thermotolerant yeast strain.

Non-TAQed cell-fused diploid strains (SNT17, CF1, CF2, CF3, CF4, CF5, CF6) were generated by cell fusion of the W700M2 and N44D strains. The cell-fused TAQed mutant was generated by TaqI activation of W700M2 and N44D followed by cell fusion.

**Plasmid construction**. TaqI was amplified from the *Thermus Thermophilus* HB8 (Takara, Shiga, Japan) genome and cloned into pYES2.1/V5-His TOPO (ThermoFisher Scientific, Waltham, MA, USA) and pORF-CLONE vectors (MoBiTec, Göttingen, Germany) using 5BamTaqI2 (5′-CGCGGATCCAAAATGGCCCCTA-CACAA-3′) and PstV5 (5′-TTTTCTGCAGCGTAGAATCGAGACCGAG-3′) primers. These plasmids express the TaqI-V5-6× His protein. The pORF-CLONE-TaqI-V5-6× His was referred to as pTaqI in this study. TaqI was also cloned into a

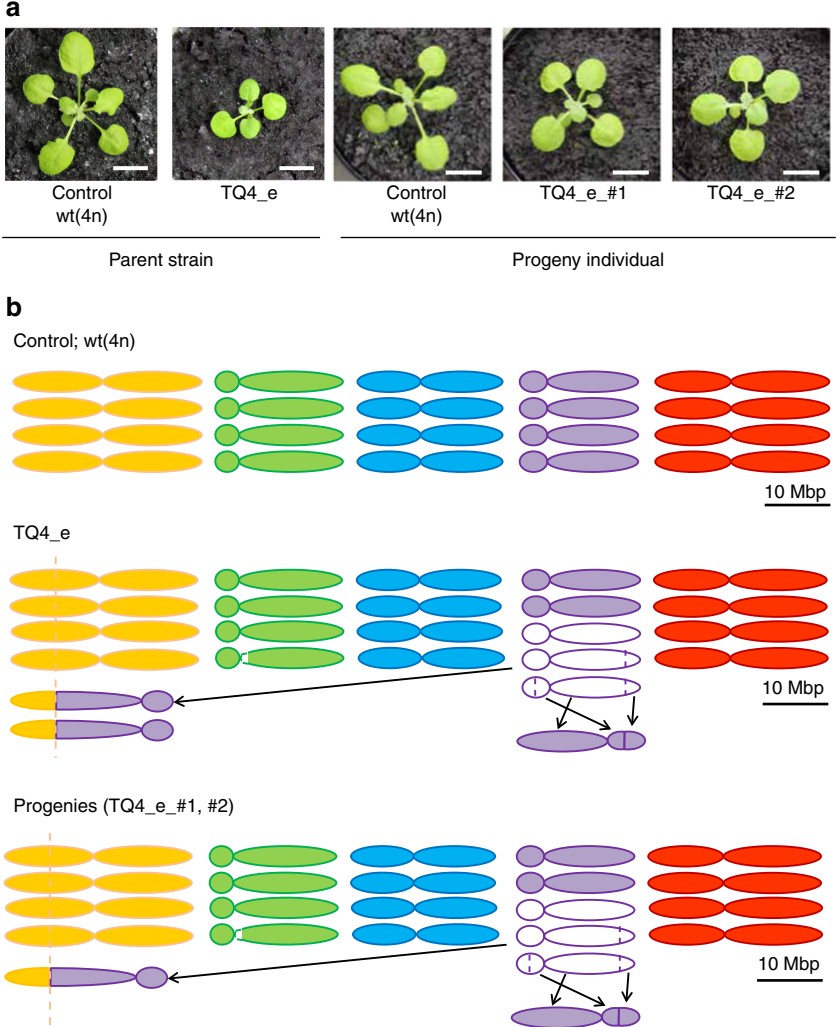

**Fig. 5** Changes in leaf size and estimated chromosome structure in TQ4_e and progenies plants. **a** Images and **b** estimated genome structure of TQ4_e (TAQed mutant), progenies (TQ4_e_#1 and #2) and control plants grown in soil for 3 weeks. Scale, 10 mm

pET-15b vector (Novogene, Chula Vista, CA, USA) using NT305 (5′-CAT-ATGGCCCCTACACAAGCCCA-3′) and NT191 (5′-GGATCCT-TACGGGCCGGTGAGGGC-3′) primers to express the 6×His-TaqI protein in *Escherichia coli*.

**Site-directed mutagenesis of TaqI**. In a previous study, the mutation of Asp[142] (D142A) abolished DNA binding and DNA cleavage in vitro[49]. Here, the template plasmid (pTaqI) was amplified with mutagenesis primers (5′-GACGCCCAGGG-GATAGCTGCACAAATTCAAGCA-3′ and 5′-TGCTTGAATTTGTGCAGC-TATCCCCTGGGCGTC-3′) and the amplicon was digested with DpnI to eliminate the template DNA for further analysis. The digested amplicons were transformed into competent cells and site-directed mutagenesis of TaqI (dTaqI) was verified by sequencing.

**TaqI activation and viability assay**. Haploid cells harboring pTaqI or the vector backbone were cultured in SD/MSG or MM at 30 °C. Upon confluency, 150 μM CuSO₄ was added to the medium and cultured for an additional 4 h to induce TaqI expression. Cells were washed with water, aliquoted, and incubated for the indicated times at 42 °C. Viability was assessed by plating on YPD or SD-uracil agar. Diploid strains were cultured in SD at 30 °C, and addition of CuSO₄ and heat treatment were performed as described.

**Assessment of HR and loss of heterozygosity assays**. The MJL1720 strain, a diploid derivative of SK1 with *arg4-bgl/arg4-nsp* heteroalleles, was used to monitor HR frequency. After TaqI activation, cells were spread on SD-arginine and YPD plates, which were examined for *ARG4*⁺ colony formation. Fused diploid WT14 cells harboring a heterozygous mutation in the *ADE2*⁺ locus were subsequently

examined for HR. Loss of heterozygosity (loss of functional *ADE2*⁺) was assessed by the presence of red (*ade2*⁻) colonies on SD plates.

**Canavanine-resistance assay**. To investigate Rev3-dependent TAQing-induced mutation frequency, we employed a canavanine-resistance assay. The arginine analog canavanine is toxic to cells, and mutations in the arginine permease gene (*CAN1*) restores this toxicity. After TaqI activation, cells were spread on SD-arginine plates in the presence or absence of 100 μg/mL canavanine. Resistant colonies were counted and compared to mock-treated controls.

**Microscopy and antibodies**. To analyze TAQed diploid strains, cells grown in YPD at 30 °C were harvested, fixed with glutaraldehyde, and washed with phosphate-buffered saline before observation. Images were acquired using an EVOS FL Cell Imaging System (Thermo Fisher Scientific) and analyzed by Adobe Photoshop CS 5.1 (Adobe Systems, San Jose, CA, USA). Rad51 immunostaining was performed as previously described[50]. Anti-*S. cerevisiae* Rad51 (62-101; BioA-cademia, Osaka, Japan) and Alexa488 goat anti-rabbit IgG were purchased from Thermo Fisher Scientific. Images were acquired using a BZ-X700 system (Keyence Corp., Osaka, Japan). Anti-FLAG (M2 F1804; Sigma-Aldrich, St. Louis, MO, USA), anti-tubulin (sc-53030; Santa Cruz Biotechnology, Dallas, TX, USA), and anti-V5 (46-0705; Invitrogen, Carlsbad, CA, USA) were used for western blot analysis. Uncropped blots can be found in Supplementary Fig. 11.

**Pulsed-field gel electrophoresis**. Chromosomal DNA embedded in 1% InCert agarose plugs (Lonza, Basel, Switzerland) were treated using zymolyase (Seikagaku-kogyo), RNase A, and proteinase K. Plugs were electrophoresed on a 1% Magabase agarose (Bio-Rad, Hercules, CA, USA) gel in 0.5× Tris/borate/EDTA buffer using CHEF Mapper (Bio-Rad). Electrophoresis was performed out at 14 °C with 6 V/cm

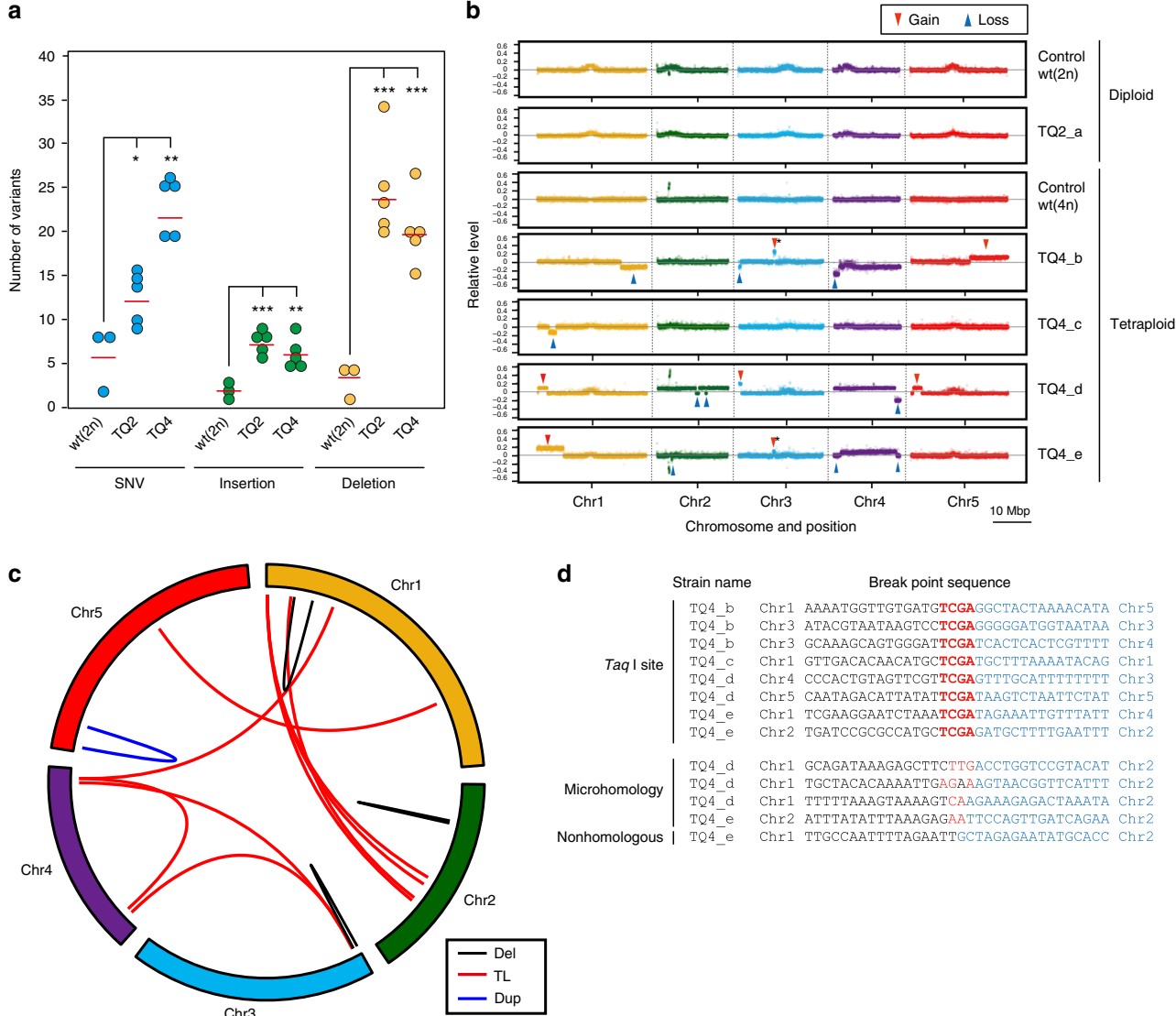

**Fig. 6** TaqI-induced large copy number variations (CNVs) and chromosome rearrangements in *Arabidopsis* plants. **a** Number of SNVs and InDels in TAQed diploid (TQ2) and tetraploid (TQ4) progenies. Data represent the number of individual variants (dots) and the mean (red line) *$P < 0.05$, **$P < 0.01$, ***$P < 0.001$; Welch's *t*-test. **b** Array comparative genome hybridization (aCGH) karyotyping of TAQed diploid (TQ2_a) and tetraploid (TQ4_b-e) *Arabidopsis* plants with large CNVs and control plants [wt(2n) and wt(4n)]. *Y* axis exhibit relative fluorescence intensity of $\log_{10}$ (Cy5[sample]/Cy3[control]). Large CNVs (gain) in the upper arm of Chr3, indicated by an asterisk, were excluded from subsequent analysis because they occurred independently in multiple progenies. **c** Circular ideogram showing all TLs and large deletions identified in four TQ4 strains. **d** Breakpoint sequences in TAQed tetraploids with large CNV events. Consensus sequences are shown in red

voltage for 24 h. The angle was 120°, and the pulse time was ranged from 60 to 120 s. The DNA was stained with SYBR Green (Lonza, Rockland, ME, USA).

**Purification of TaqI from *E. coli*.** *E. coli* BL21 (DE3) pLysS cells were used to produce recombinant TaqI. Expression of 6× His-tagged TaqI was induced for 24 h at 18 °C by addition of 0.1 mM isopropyl β-D-1-thiogalactopyranoside (IPTG). Extracts were prepared by sonication and TaqI was purified using a TALON Metal Affinity Resin (Clonetech, Mountain View, CA, USA). The purified protein was then concentrated using a 10 kD cut-off filter (Millipore, Billerica, MA, USA).

**TL rearrangement-distance estimation from Hi-C data.** To determine the closeness of TL-junction sites are close in nuclear space, 3D-coordinate information of yeast chromosomes was obtained from a previous study[22], and 10,000 pairs of points on the chromosomes were randomly selected using Matlab_R2017b (https://www.mathworks.com/) to estimate the approximate distances between any pairs of points and plotted in a histogram (considering a ~2 μm nuclear size for the diameter). The distances between pairs of breakpoints at the sites of TL events, including six non-homologous junctions and two Ty-like homologous junctions, were also calculated from the 3D

coordinates. Based on the random distance distribution, the probability of a certain distance was estimated, and each TL-junction distance was plotted. The appearance of pairs of breakpoints in the 3D-coordinate plane was plotted in the 3D space using Matlab.

**Growth-rate measurement and ethanol production mass.** Cells from strains W700M2, N44D, non-TAQed strain SNT17, and the TAQed mutant SNT11-2×40 were pre-cultured in SD (6.7 g/L yeast nitrogen base with all amino acids and 20 g/ L glucose) at 30 °C. Cells were then shifted to 5 mL of test media SD or SX (6.7 g/ L yeast nitrogen base with all amino acids and 20 g/L xylose) at the very low concentrations in test tubes for culturing at 30 or 40 °C while measuring the rate of transmitted light (950 nm) volume at 1-min intervals using an ODBox-C/ ODMonitor system (Taitec Corp., Saitama, Japan). The transmission rate was converted into optical density (OD), and the average OD over 20 min was calculated for each sample. At the time points of 8, 24, and 48 h from the medium change, 250 μL of the samples was harvested to measure ethanol concentration using an F kit-ethanol (J.K. International, Tokyo, Japan) according to the manufacturer instructions.

**Table 2 Frequency of large CNV in TQ4 plants**

| | TQ2 ($n = 12$) | TQ4 ($n = 80$) | Control wt(4n) ($n = 16$) |
|---|---|---|---|
| Large CNV (number of plants) | 0 | 22 (9) | 0 |
| Large CNV rate (CNV event per plant) | 0 | 0.275* | 0 |
| Chromosomal CNV (number of plants) | 0 | 42 (37) | 3 (3) |
| Chromosomal CNV rate (CNV event per plant) | 0 | 0.53* | 0.19 |

*$P < 0.05$ by Welch's $t$-test

**DNA extraction and tiling-array analysis**. The genomes of haploid wild-type, cell-fusion, and mutant strains were analyzed using a tiling array (Agilent Technologies, Santa Clara, CA, USA). Genomic DNA was extracted using spin columns (Genomic DNA Buffer Set No. 19060, and Genomic tip 500/G No. 10262; Qiagen, Hilden, Germany). *A. thaliana* genomic DNA was extracted using spin columns (DNeasy Plant Mini Kit No. 69104; Qiagen). For tiling-array analysis, array comparative genomic hybridization (aCGH) was performed using a *S. cerevisiae* or *A. thaliana* custom microarray (designed by eArray; Agilent Technologies) for genome-wide copy number analysis. Genomic DNA (100 ng) was fluorescently labeled using the SureTag DNA Labeling Kit (Agilent Technologies) according to the manufacturer's instructions. Prior to hybridization, sample and control probes were incubated with the blocking agent and hybridization buffer contained in the Oligo aCGH/ChIP-on-chip Hybridization Kit (Agilent Technologies) at 95 °C for 3 min followed by 37 °C for 3 min. Hybridization was performed at 65 °C for 24 h in a microarray hybridization oven (Agilent Technologies) according to the manufacturer instructions, and the slides were washed with the Oligo aCGH/ChIP Wash Buffer Kit (Agilent Technologies). Array data were quantified and normalized using Feature Extraction Software (v.11.0.1.1; Agilent Technologies).

**Genome sequencing**. Haploid YPH499 and S799 strains were sequenced using PacBio RS II (Pacific Biosciences) and MiSeq (Illumina, 2 × 300 bp; Illumina) systems. Cell-fusion diploid strains were sequenced using MiSeq (2 × 300 bp; Illumina). PacBio 20-kb SMRTbell libraries were generated by using the SMRTbell template prep kit 1.0 and BluePippin size selection systems (Pacific Biosciences) according to manufacturer instructions. The libraries were sequenced on three SMRT cells (one for YPH499 and two for S799). MiSeq libraries were generated using NEBNext Ultra DNA library prep kit for Illumina (E7370S; New England Biolabs, Ipswich, MA, USA) and the NEBNext multiplex oligonucleotides for Illumina (E7335S; New England Biolabs). Genome sequencing of TAQed plant genomes were examined by a commercial company (Hokkaido System Science) using an Illumina HiSeq 2500 (Illumina). Preparations for the genomic DNA library (350 bp inserts) were constructed using the TruSeq Nano DNA LT sample prep kit (Illumina) according to the manufacturer standard protocol.

**Computational analysis**. Analysis methods for variant detection was differed for *S. cerevisiae* and *A. thaliana*. For yeast, genomic sequences of parental haploid strains were determined as the reference sequences for mapping and variant detection. The sequenced reads of TAQed strain genome were mapped using a Burrows–Wheeler aligner (BWA)[51], and small variants were called using the Genome Analysis ToolKit (GATK)[52]. For *Arabidopsis*, the genome sequence uploaded onto the TAIR10 database (https://www.arabidopsis.org/) was used as the reference sequence for mapping and variant detection. Furthermore, sequenced reads of TAQed plant genome were mapped using commercial software (CLC Genomics Workbench v.8.5; Qiagen). These variants were also called using this software, with variant-detection workflow shown in Supplementary Fig. 12.

**Yeast reference-genome sequencing**. To detect the rearrangements induced by the TAQing system, genomic sequences of parental haploid strains were analyzed. Bioinformatics workflow for reference-genome construction is shown in Supplementary Fig. 13. PacBio sequencing data for the two parental haploid wild-type strains were filtered and sub-reads were generated (Supplementary Fig. 13 and Supplementary Data 3) using SMRT Analysis v.2.3.0 (Pacific Biosciences) and smrtpipe.py (v1.87.139483) with the default parameters. De novo assembly was conducted using Sprai[53] (v.0.9.9.14) with the default parameters. Newly assembled contigs were roughly equivalent to those of the widely used laboratory strain S288C (previously reported chromosomes I–XVI and ChrM; obtained from the Saccharomyces Genome Database[54]), except chromosome XII. Because chromosome XII harbors up to 200 copies of rDNA at ~10 kb in length, the contigs with partial chromosome XII sequences were separated into three or more contigs [left arm, the right arm, and the rDNA region(s) in both strains]. These consensus contigs were partially corrected using the paired-end short-read sequencing data from MiSeq (Illumina). For error correction, the MiSeq data were mapped to the backbone of the assembled contigs using BWA, and then the SNPs and InDels in the backbone contigs were detected and modified by Picard (http://broadinstitute.github.io/picard/), GATK, and perl scripts. The contigs were further improved using the gap-

closing module in PBJelly[55] (PBSuite 15.8.24), and the sequences were corrected once again with the MiSeq data as described above. Each of the contigs from YPH499 and S799 were sorted and renamed after S288C annotation using BLAST 2.2.29+[56]. For each chromosome, centromeres with sequences identical to those of S288C were marked using EMBOSS[57]. Two haploid genome references were combined for use as the reference genome of the "diploid" cell-fusion strains.

**Variant detection in haploid genome references**. SNPs and InDels between the S799 and YPH499, YPH499, and S288C strains were detected, and their distributions in the respective genomes, as well as between the two strains, were determined. SNPs and InDels between S799 and YPH499 were widely spread along the entire chromosome (average 6.5 SNPs/kb and 0.7 InDels/kb; ~0.72% difference; Supplementary Fig. 14) and sufficient to distinguish two different alleles, except for those in the rDNA regions of chromosome XII and some telomeric regions, using 300-bp × 2 paired-end Illumina sequencing. YPH499 derived from S288C is highly conserved with its ancestor, S288C (a ~0.05% difference; Supplementary Fig. 14). SNP and InDel detection was performed as follows. First, haploid MiSeq sequencing data for S799 was aligned with that of the YPH499 reference genome using BWA. Additionally, haploid YPH499 data was aligned with that of the S288C reference genome. Then PCR duplicates were then removed using Picard (http://broadinstitute.github.io/picard/), and raw variants were called using GATK (for SNP parameters: QD < 2.0, FS > 60.0, MQ < 40.0, MQRankSum < −12.5, ReadPosRankSum < −8.0, SOR > 4.0 and InDel parameters; QD < 2.0, FS > 200.0, ReadPosRankSum < −20.0, SOR > 10.0) to recalibrate base quality. Variants were called again with the same filtering parameters and chromosome-wide variant distribution was plotted at every 20 kb (Supplementary Fig. 15).

**Mutation and rearrangement detection**. MiSeq reads of the cell-fusion (control) strain and 13 TAQed strains were mapped using BWA. Both whole-genome sequencing and tiling-array data were used to detect large-scale genome rearrangements and aneuploidies (Supplementary Fig. 16). Chromosome-wide CNVs detected by both sequencing and tiling array were defined as "aneuploidy". Chromosomal rearrangements with the junction of different segments of chromosomes were defined as "translocation (TL)." For the TL detection, the local CNVs of the segemnts were also confirmed using the tiling-array data. The corresponding breakpoints were detected by extracting chimeric reads around the junction sites, which were mostly non-homologous; however, three homologous-mutation points were detected. These junction sequences were further analyzed by BLAST to determine whether they were derived from Ty-like repeat sequences or rDNA regions. These TL sites except for the TL between rDNA regions were verified by PCR (local amplification of ~5 kb across the junctions). Chimeric chromosomes of homologous sequences from S799 and YPH499 and exhibiting changes in partial coverage or signal intensity in both sequencing and tiling-array data were defined as "Break Induced Repairs (BIRs)". Chimeric chromosomal regions (~0.7–30 kb) in S799 and YPH499 and exhibiting changes in partial coverage detected in the sequencing data were referred to as "Short Gene Conversions (SGCs)". To search for BIR and SGC, zero-coverage regions on one of the references were marked as candidate regions to test whether their homologous chromosome pair displayed double-coverage reads[58]. If each end of the chromosome was derived from both S799 and YPH499, the mutation was annotated as a BIR, and rearranged positions were further searched from the chimera sequences around the events using BLAST, samtools (http://samtools.sourceforge.net/), and perl/shell scripts. Partial chimeric sequences from the pair within a chromosome were annotated as SGCs. Because homologous TLs, BIRs, and SGCs represent homologous rearrangements, approximate rearrangement positions were notated. Small variants were detected using Picard and GATK. SNVs and InDels reported in AT-rich regions, rDNA regions, telomeres, and regions with too low-coverage (<50% of the average coverage) and that are often seen especially around the large chromosomal rearrangements were eliminated as false positives[59]. Common SNVs and InDels detected in all control and TAQed strains were also eliminated as false positives. Genome-rearrangement/mutation events are visualized in Figs. 2a and 6c as circular plots generated by the R package (ggbio; https://bioconductor.org/packages/release/bioc/html/ggbio.html)[60].

**Re-sequencing analysis in *A. thaliana*.** All Illumina reads from control and TAQed plants were trimmed using CLC Genomics Workbench (v8.5; Qiagen) to a minimum quality score of 0.01 (equivalent to a Phred quality score of 20), a minimum length of 95 bp, and adaptor sequences were removed. The following parameters were changed from default: Ambiguous trim = Yes, Quality trim = Yes, Quality limit = 0.01, search on reversed sequence = Yes, Remove 5′ terminal nucleotides = No, Minimum number of nucleotides in reads = 95, discard short reads = Yes, Remove 3′ terminal nucleotides = No. For mapping and variant calls, filtered reads were mapped to the reference (TAIR10; https://www.arabidopsis.org/) using the "Map Reads to Reference" tool in the CLC Genomics Workbench (v8.5; Qiagen). The following parameters were changed from default: Masking mode = No masking, Mismatch cost = 2, Insertion cost = 3, Deletion cost = 3, length fraction = 0.5, Similarity fraction = 0.8, Global alignment = No, Auto-detect paired distances = yes, Non-specific match handling = Ignore, Collect unmapped reads = No. Variant calling was performed on a read mapping using the "Basic Variant Detection" tool in the CLC Genomics Workbench (v8.5; Qiagen). The optional parameters are shown below; Ignore positions with coverage above = 100,000, Restrict calling to target regions = Not set, Ignore broken pairs = Yes, Ignore non-specific matches = Reads, Minimum coverage = 10, Minimum count = 2, Minimum frequency (%) = 35.0, Base quality filter = No, Read direction filter = No, Read position filter = No, Relative read direction filter = yes, Significance (%) = 1.0, Remove pyro-error variants = No. Based on variant-call information, mapped reads were locally realigned using the "Local Realignment" tool in the CLC Genomics Workbench (v8.5; Qiagen). The optional parameters are shown below; Realign unaligned end = Yes, Multi-pass realignment = 2, Force realignment to guidance-variants = No. Detection of InDels and structural variants was performed on locally realigned read mappings using the "InDels and Structural Variants" tool in the CLC Genomics Workbench (v8.5; Qiagen). The optional parameters are shown below; P-value threshold = 0.0001, Maximum number of mismatches = 3, Minimum number of reads = 2. Based on InDel and structural-variant information, mapped reads were locally realigned using the "Local Realignment" tool in the CLC Genomics Workbench (v8.5; Qiagen). The following parameters were changed from default: Force realignment to guidance-variants = Yes. Detection of small variants, InDels, and structural variants was performed on a locally realigned read mapping using the "Basic Variant Detection" and the "InDels and Structural Variants" tools in the CLC Genomics Workbench (v8.5; Qiagen).

**Verification of small variants.** After local realignment, small variants were called using the Basic Variant Detection tool in CLC Genomics Workbench (v8.5; Qiagen). To narrow small-variant candidates in TAQed strains, those detected in control plants were excluded. Variants meeting the criteria of mutation frequency (mutated read count/total read count > 0.4) were selected and manually verified. For SNVs, regions harboring variants on only one locus were excluded. InDels were selected based on two criteria: (1) <2 reads containing InDels in control plants and (2) InDel uniqueness.

**Validation of structural variants.** To narrow the list of called structural variants, those detected in control strains were excluded. We adopted three-step strategies to verify structural variants (Supplementary Fig. 17). In the first step, CNV mapping results were compared with tiling-array data to verify the CNV region, and if they were not included in the list of structural variants, they were added as candidates. In the second step, soft-clipped sequences were extracted on the boundary line of CNVs (boundary region A) where partially unmapped reads (>10% reads of total reads were partially unmapped) were highly accumulated. The soft-clipped sequences were analyzed by BLAST against the TAIR10 database (https://www.arabidopsis.org/), and those at the site of the best ($e$ < 1E−04) BLAST result were extracted (boundary region B). Extracted sequences were analyzed by BLAST against the reference genome, and if the site of the best BLAST result was region A, it was considered a reciprocal best hit. For the third step, SAM files were analyzed, and accumulated discordant pair-end reads surrounding regions A and B were mapped onto boundary regions A and B, respectively. Alignment start position for the pair reads was subsequently analyzed, with these results agreeing with BLAST searches of soft-clipped sequences.

**Plant materials and growth conditions.** The *A. thaliana* GU–US reporter (direct repeat-type) line 1406 has been previously described[21,61]. Ecotype Col was used as a control strain. Plants were grown in corner dishes on germination medium (MS medium containing 1% sucrose and 0.8% agar) for 3 weeks with a 16-h/8-h light/dark cycle. To activate TaqI, 1-week-old plants were grown at 37 °C for 24 h. All plant strains are listed in Supplementary Data 2.

**Construction of transgenic plants.** The 35S:TaqI-NLS (NLS; nuclear localization signal) plasmid was constructed by cloning PCR-amplified fragments. The primers BamHI-TaqI-F and TaqI-SacI-R were used to amplify the TaqI fragment while BamHI-TaqI-F and TaqI-NLS-SacI-R were used to add the NLS tag. The amplified sequence (TaqI-NLS) was ligated into the pCR2.1 vector with the TOPO TA Cloning Kit (Invitrogen). The TaqI-NLS sequence was inserted behind the 35S

promoter in the pBI121 plasmid to generate the plant transformation vector, which was introduced into the Arabidopsis line 1406 by *Agrobacterium*-mediated transformation using the floral dip method[62]. TaqI-NLS transgenic plants were selecting by growing T1 seeds on plates containing MS medium supplemented with 50 mg/L kanamycin. T3 progeny of homozygous plants harboring a single copy of the transgene were selected [TaqI$^+$(2n)]. The following primers were used for cloning: BamHI-TaqI-F (5′-AGGATCCCCGGGTGGTCAGTCCCTTATGGCCCCTACA-CAAGCCCA-3′), TaqI-SacI-R (5′-AGAGCTCTGTACCTCACGGGCCGGTGAG GGC-3′), and TaqI-NLS-SacI-R (5′-AGAGCTCCCCGGGCTATCCTCCAACCT TTCTCTTCTTCTTAGGCTGCAGACCTCCCGGGCCGGTGAGGGCTTCC-3′).

**Generation of polyploid plants.** Three-week-old plants [control and TaqI$^+$(2n)] were dipped in a 0.01% colchicine solution for 3 min, and progeny seeds were obtained by self-propagation. The nuclear phase of progeny plants was analyzed by flow cytometry on a Cell Lab Quanta SC MPL (Beckman Coulter, Brea, CA, USA) using a mercury arc lamp (365 nm), a UG1 excitation filter, a 560 dichroic short-pass filter, and a 450/55 bandpass filter. Nuclear extraction and DNA staining of nuclear DNA from *Arabidopsis* tissue were executed with CyStain UV Precise P (Sysmex Corp., Kobe, Japan). Tetraploid control plants (1406P4) and tetraploid 35S:TaqI plants [TaqI$^+$(4n)] were selected.

**RNA analyses.** Total RNA was extracted from the leaves or roots of 1-week-old *Arabidopsis* plants grown on MS agar plates using an RNeasy Plant Mini kit (Qiagen) according to manufacturer protocol. For quantitative reverse-transcription (qRT)PCR analysis, cDNA was synthesized using a high-capacity cDNA reverse transcription kit (Applied Biosystems) with random primers and according to manufacturer instructions. qRT-PCR was performed on an ABI 7000 real-time PCR system (Applied Biosystems) using Power SYBR Green PCR Master Mix (Applied Biosystems) and the following primers: 18SrRNA-F (5′-CGGCTAC CACATCCAAGGAA-3′), 18SrRNA-R (5′-TGTCACTACCTCCCCGTGTCA-3′), AtBRCA1-F (5′-CCATGTATTTTGCAATGCGTG-3′), and AtBRCA1-R (5′-TGT GGAGCACCTCGAATCTCT -3′).

**Recombination assays.** For recombination assays, strains 1406 and 1406P4 served as controls, and histochemical GUS staining was performed as described previously[63]. The number of GUS blue spots (indicative of a recombination event) on each cotyledon was determined visually under an optical microscope (Olympus). For each line, 30 to 100 cotyledons were analyzed.

**Data availability.** Sequencing data are deposited in the DDBJ Sequence Read Archive database under accession numbers DRA005252 and DRA005268. Micro-array data are deposited in the NCBI GEO under accession number GSE90027 (SuperSeries). This SuperSeries is composed of the following SubSeries: GSE90025 and GSE90026.

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

## Acknowledgements

We thank S. Ohsato, H. Seo, K. Hirota, Y. Hosono, K. Arai, H. Sasaoka, S. Mura, and E. Takaya, for their valuable discussion; K. Tokuhiro, and S. Katahira for xylose-utilizable and a thermotolerant yeast strains construction; R. Kitagawa, J. Mushika, R. Nagae, T. Tanaka and R. Nakamura for their technical support; B. Hohn, S. Toki, M. Endo for the generous gift of Arabidopsis 1406 strain seeds; S. Koide, K. Kawano, Y. Tadokoro for assistance with bioinformatics analysis. We thank Editage (www.editage.jp) for English language editing. This work was supported by a Grant-in-Aid for basic science (17H03711) and for Scientific Research on Innovative Areas (23114003) and Platform for Dynamic Approaches to Living System from the Ministry of Education, Culture, Sports, Science and Technology, Japan (to K.O.). K.O. is also supported by the Basic Science and Platform Technology Program for Innovative Biological Medicine from the Japan Agency for Medical Research and Development (AMED). Computational resources were provided by the Data Integration and Analysis Facility, National Institute for Basic Biology.

## Author contributions

K.O., T.S., and N.Mi. conceived the study. A.O., H.T., K.K., and N.Mu. designed and performed the genomic analyses. N.Mu., H.T., and A.I. performed the aCGH array analysis. T.N., A.O., K.K., K.S., A.K., S.Y., and A.I. designed and constructed the yeast strains and mutants and performed the biochemical analyses. N.Mu., H.T., H.S., S.K., and C.O. generated the transgenic *Arabidopsis* strains, and derived mutants and performed physiological analyses. All experiments were performed under the supervision of K.O., N. Mi., and/or N.Mu. K.O., N.Mu., A.O., H.T., and T.N. wrote the manuscript, and all authors read and commented on the manuscript.

## Additional information

**Competing interests:** This work was supported by Toyota Central R&D Labs., Inc. (Japan) and Toyota Motor Corporation (Japan). N.Mu., H.T., A.I., H.S. and N.Mi. are employees of Toyota Central R&D Labs., Inc. S.K. and C.O. are employees of Toyota Motor Corporation. Patent applications have been filed for the technology described in this publication. N.Mu., H.T., T.N., K.K., H.S., S.K., C.O., T.S., N.Mi. and K.O. are named as the inventors of these patents. The remaining authors declare no competing interests.

