## [Peer Review File · Nature Communications]

Reviewers' Comments:

Reviewer #1:

Remarks to the Author:

In the manuscript "Monitoring the genome reorganization process after genome duplication revealed substantial phenotypic effects" the authors describe a system to generate chromosomal rearrangements by inducing genome-wide DNA double strand breaks (DSB) with the restriction endonuclease TaqI. The authors suggest that 'TAQing' is better tolerated in cells with higher ploidy, both in *S. cerevisiae* and *Arabidopsis*, and can produce phenotypic diversification. Additionally, the authors sequence genomes derived from TAQing to investigate the mechanisms by which DNA repair occurs following transient induction of genome-wide DSBs.

Overall I think that TAQing is a really interesting idea and a tool that could be broadly useful. For instance, TAQing could be used as a tool in metabolic engineering to generate new genomic sequences that improve pathway flux in microbial and plant production strains. Additionally, TAQing could be used as a tool to study aspects of genome chaos, gene networks, or potentially to generate minimal genomes in a completely unbiased manner and under different growth conditions.

Here the authors present TAQing primarily as a model for genome re-organization after a whole genome duplication event (WGD). Below I outline major and minor comments. With major revisions, in principle I am supportive of publication in Nature Communications and would be happy to review the manuscript again.

Major comments

1. The title and abstract do not accurately reflect the contents of the manuscript. For instance the title doesn't mention that the genome reorganization process is achieved via induced double strand breaks. Further, there is no mention of TaqI in the abstract. Both the title and abstract should be modified extensively.
2. Whether the simultaneous induction of ~200 DSB per cell at a defined sequence leading to genome rearrangement is an appropriate model for studying the downstream effects of WGD is a hard question to answer. The authors address shortcomings of their approach in the discussion.
3. More controls should be shown to establish the viability of yeast cells post TAQing. In Figure 1E, viability should be quantified for all conditions that could contribute to lethality and should include (i) no Cu²⁺/30°C (mock); (ii) Cu²⁺/30°C; (iii) no Cu²⁺/42°C; (iv) Cu²⁺/42°C. Also, a TaqI endonuclease-dead mutant could be used to distinguish lethal genome rearrangements from lethality due to heat shock/copper.
4. It isn't clear from the manuscript text or the figure legend whether the data in Figure 1E is from a haploid or diploid cell. In either case, viability should be quantified for both haploid and diploid cells. Presumably viability would be higher for the diploid cells and would provide additional support for the conclusion that TAQing is better tolerated by diploid cells.
5. When quantifying viability, after TAQing cells should be plated on medium that selects for the TAQ plasmid rather than YPD. There is a huge selection pressure favoring cells that have lost the plasmid.
6. For the yeast TAQing experiments, the authors should clearly state how many colonies post-TAQing

were evaluated for genome rearrangements, by what method were they evaluated, and how many actually have rearrangements.

7. Throughout the manuscript the authors refer to 'mock' versus TaqI activation. Please modify figure legends to clarify that the mock condition – for instance, for yeast I assume this means without Cu²⁺ at 30°C.

8. Why wasn't TAQing carried out in a tetraploid yeast cell? Isn't this a better model of WGD?

9. Lines 95-96 - it's not clear how the pulsed-field gel provides evidence to support this conclusion. Please clarify.

10. In the discussion, the authors should reference and compare TAQing to SCRaMbLE (PMID: 265666580), which is another means by which to generate genome rearrangements in yeast.

11. In principle, chromosomes that are close in nuclear space are more likely to undergo translocations as a function of TAQing. Have the authors attempted to correlate Hi-C data with TAQ induced genome rearrangements?

Minor comments

1. Figure 1 legend – include the strain number for each experiment. It is difficult to tell whether each experiment is carried out in a haploid or diploid cell for panels (d) and (e)

2. Figure 1A – the homologous chromosomes should be the same size.

3. Line 36 – should read "we have established a new tool that induces DNA double strand breaks and extensively...".

4. Line 105 – should read "diploid cells can undergo gross chromosomal rearrangements at a higher frequency than haploid cells."

5. Line 127 – define SGD

6. Line 128-129 – isn't templated repair exactly the benefit of having a second chromosome following a DSB?

7. Line 129 – define TL

8. Line 131 & 215 – I don't think the term microhomology is applicable here. Should be sticky end?

9. Panel labels on the figures/figure layout are hard to follow. Should be rearranged so they appear in more logical order.

10. Line 171 – define GUS

11. The authors should include a strain table in the supplement with all parental strains as well as strain numbers for all analyzed strains in the manuscript (eg. strains produced by TAQing).

12. Plasmids for TAQing should be contributed to Addgene.

Reviewer #2:

Remarks to the Author:

Understanding the mechanisms of chromosome rearrangements is an exciting and important area of research. It is well accepted that whole genome duplication (WGD) promotes large-scale genome rearrangements that provide raw materials for adaptive evolution. The problem, however, is that this idea relies on the genomic analysis of the past WGD-experienced species, and it remains poorly understood what exactly happens after WGD, how rearrangements occur, and whether these rearrangements can contribute to evolution. The study by Muramoto et al investigates chromosomal rearrangements induced in yeast and Arabidopsis following WGD. In particular, they established a very

powerful and innovative tool to induce gross chromosomal rearrangements (GCRs). GCRs are induced by introducing simultaneous genomic DNA double strand breaks (DSBs), which are formed at one time in living cells by the heat-activated endonuclease Taq1. They applied this tool to yeast and Arabidopsis cells with different ploidy, and concluded that yeast and plants with larger ploidy can undergo much more complex genome rearrangements. In addition, the authors report that genome-rearranged yeast and plant cells showed enhanced phenotypic diversity. Overall, the authors propose that genome duplication allows more complex genome rearrangements and phenotypic changes that are promoting adaptive evolution. Overall, this new approach to the analysis of GCRs, their formation, and connection with WGD and evolution represents an important development in the field of genetic stability and evolution, and the results of this study are of interest for researchers working in many areas, including DNA repair, recombination, and evolution. The side-by-side comparison of yeast and plants and of organisms with different ploidy combined with detailed characterization of mutations and genome rearrangements genome-wide are among the main strengths of this work. Because interest in this topic is very high, I expect that this paper will be frequently cited and this approach will be used in various areas.

Specific comments:

1. P.2. Abstract.

"We found large-scale rearrangements occur frequently in diploid yeasts and tetraploid plants, whereas haploid yeasts and diploid plants were little rearranged". This is the most important conclusion of the paper, and it is definitely supported by data and exciting. However, it is important to compare, side-by-side, cell viability and the number of breaks produced by Taq1 in yeast haploids vs diploids and in plants. I was able to find (Fig. 1e) that Taq1 induction leads to the lethality of at least 40% of cells in haploids. Is it possible that all rearrangements occurred in this fraction? How does it compare with viability in diploids? The lethality of GCRs in haploids is the simplest explanation of the main result, and it will be important to provide all of the necessary comparisons. Also, is it that the number of breaks in diploids is approximately 2x of that in haploids?

2. Fig. 2g and p. 5, line 101. "In addition, heat activation markedly induced gross chromosomal rearrangements". Fig. 2g represents loss of heterozygosity events. Why are they chromosomal rearrangements if they result from exchange between homologs?

3. P. 6, line 111. "Genomic alterations induced by activated Taq1 were further examined using strains with altered morphology ("mutants") by combined long/short high throughput sequencing". From this statement it appears that the mutants were selected. It will be helpful to describe the process of selection further. In particular, it is important to explain how the frequencies of chromosomal rearrangements observed in this experiment characterize the frequency of rearrangements in the entire population.

4. P. 6, line 120. "These mutations likely resulted from error-prone DSB repair consistent with a previous report". The authors propose that mutations result from the mechanism that was proposed to result from the damage accumulated in ssDNA resulting from the DSB resection. Do the authors believe that these mutations were associated with repair of Taq-induced breaks that proceeded not via NHEJ, but via HR or MMEJ associated with long resection? Some discussion will be very useful here.

5. P. 7, line 127. The term SGC needs explanation. I was able to find its meaning only in figure legend.

6. P.7, line 128. "This is surprising since homologues are assumed to exist independently in vegetative yeast nuclei". I am not sure why it is surprising. Gene conversion, especially allelic gene

conversion is very efficient in yeast and does not require homologues to stay together all the time.

7. P. 7, line 129. " Detectable TL breakpoints were observed in 66.6% (6/9) of Taq1 recognition sequences". This requires clarification. Do the authors mean all Taq1 recognition sequences around the genome? Or do they mean that every breakpoint correlated with a Taq1 recognition sequence?

8. P. 7, line 135. "Two isolates harbored chromosomes with fused TL breakpoints with Ty elements, and one isolate had a chromosome fusion with a repetitive rDNA region. This is intriguing, as a previous study suggested that TLs inoriginate from HR between Ty elements or repetitive sequences". Could the authors clarify their statement: "this is intriguing"? Do they want to say that their finding contradicts those previous reports? It looks like they found the same thing. Could they please explain or re-formulate?

9. P. 36, Fig. 2d. How was mutation rate calculated? Was it calculated per generation? How many times was the experiment repeated? How were statistics for this experiments done?

10. P. 36 Fig.2e legend: "mutated chromosome": does it mean "rearranged chromosome"?

Reviewer #3:

Remarks to the Author:

The authors of "Monitoring the genome reorganization process after genome duplication revealed substantial phenotypic effects" describe a method where they induce double strand breaks given an enzyme. Furthermore, they investigate the result of these induced breaks and measure the influence of a whole genome duplication effect. The introduction, result and discussions are well written and easy to follow. They successfully test the activation of their enzyme using a heat response and quantify translocations and duplication events between different mutants and the mock cell line.

In the following I list my concerns and questions about specific topics:

1. Some sections especially in the Methods are not well written and thus hard to understand. I would advise to check and correct the text.

2. Why does the Taq1 produce more stable genotypes? As far as I understood it is a cutter enzyme. Especially for the yeast data set you see only a fraction of the cells show some phenotypic changes, right? Have you assessed other methods to induce DSB? Furthermore, I had troubles to identify the method with what the mock samples were treated.

3. You mention for Arabidopsis that the phenotype changes (leave sizes) were successfully transmitted, but leave out at what rate or over how many generations. I think that is important for your claim that changes induced are stable over the inheritance and especially "over many generations". Also for the yeast data the replication numbers that support your claim are not high and thus not 100% conclusive.

4. Please be more precise in e.g. "some isolates exhibited different..." (Page 8)

5. You need to improve the explanation of the computational methods and parameters used for the sequence analysis. For example, I had the impression that the analysis was different for the yeast vs. Arabidopsis samples. If this is the case, then please make it clear. When did you use which genome (e.g. multiple yeast strains were assembled)? You are also sometimes mentioning that you used CLC

and sometimes you seemed to use more direct methods e.g. BWA. I would furthermore recommend to cluster the subparts of the methods section according to field. E.g. all computational method description should be grouped together to enable an easier understanding of what was done.

6. Please consider rewriting the section "Validation of structural variants". I don't understand "accumulated unaligned reads (>2 reads) on the boundary line of CNV". How can you observe that these reads were unaligned, but belong to a certain region? Furthermore, 2 reads are not very stringed filter. Did you carry out any additional analysis for smaller variations or translocations?

7. For the translocations that you observe do you have validation data? I don't see that you mention this. Especially these translocations between repetitive elements are often error prone depending on the analysis.

8. I could not find the assembly statistics to see the "high quality reference genome assembly". What does "highly consistent with previous reported S288C genome sequence" mean? I would have liked to see some numbers of SNPS, alignment statistic, etc. Furthermore, you are writing "N50 contig's were partially corrected". I don't know what this means. N50 is a size measure, but you don't give any numbers.

9. For the "Mutation detection analysis" What method was used to detect the mutations? What method did you use to identify CNVs? I am a bit confused as to what you refer to translocations and CNVs. These are two or more completely different types. Translocations are most often referred as connections between two chromosomes. Whereas CNVs are alterations in the coverage (most often duplications and deletions). It can be that some translocations explain e.g. duplications for example in cancer. But they are still different types. Please clarify: " Chromosomal rearrangements for which the CNVs were also confirmed by the tiling array data were defined as "translocation," and the corresponding mutation points were detected by extracting the chimeric reads around the junction sites" (Page 25).

10. Some acronyms are hard to find (e.g. only mentioned in the figure legend: SGC)

Reviewer #4:

Remarks to the Author:

The main claim of the paper is that it is poorly understood what happens immediately after polyploidization with respect to genome reorganization and its potential impact on phenotype. This reviewer agrees with the authors that it is important to do experimental evolution where one starts with a diploid plant (or haploid yeast) and then induce various mutations (SNPs, translocations, aneuploidy, auto-polyploidy, and allopolyploidy) to see what impact these would have on phenotype. The authors have also developed a novel method to generate genome structural variation that works in plants and yeasts.

However, this reviewer has two major concerns with the paper.

First, it is false for the authors to claim novelty that they are the first to perform experimental evolution to make and watch the consequences of polyploidy from known parents.

Second, the major conclusion that polyploids have more evolutionary space for genome rearrangements and these have a major impact on phenotype is not novel; indeed, it is absolutely expected and it would be a surprise at this point to find the opposite. The authors are simply unaware

of this literature (or chose not to cite it in this paper). Just three references from one species (*Brassica napus*, or canola) are given here:

*Gaeta et al. 2007. Genomic changes in resynthesized *Brassica napus*: The effect of genomic changes on gene expression and phenotypic variation. *The Plant Cell* 19: 3403-3417

*Xiong et al. 2011. Homoeologous shuffling and chromosome compensation maintain genome balance in resynthesized allopolyploid *Brassica napus*. *Proceedings of the National Academy of Sciences, U.S.A.* 108: 7908-7913.

*Chalhoub et al. 2014. 2014. Early allopolyploid evolution in the post-neolithic *Brassica napus* oilseed genome. *Science* 345: 950-953.

The last paper includes both natural and resynthesized polyploids.

One can also find similar works on resynthesized polyploids and their major rearrangements and effects on phenotype in the literature in *Arabidopsis* (Luca Comai, Jeff Chen, Andreas Madlung labs), maize (Jim Birchler lab), wheat (Bao Liu's lab) and *Tragopogon* (Pam and Doug Soltis lab). Similarly, there has been a lot of work done in resynthesized yeast polyploids (Sally Otto and other labs). So it is no surprise the authors find their approach and major finding novel if they are not familiar with this vast body of literature.

Second, the authors methodology for examining genome reorganization is flawed in that they do not distinguish gains and losses from homoeologous exchanges which may do both simultaneously (Figures 5b, 6b, 6c, Supplemental Figures 7b, 8, 9b, 10). In the yeast case some of these may be correctly referred to as gene conversions; however, in the tetraploid plant case this may not be so. There is also the possibility of hidden aneuploidy (where one homoeologous chromosome is completely lost and replaced by its partner, although the results do not seem to show that). Some of the references given above perform these analyses and discuss these points.

In summary, this reviewer finds this manuscript deeply flawed with respect to making claims about novel experimental design ("it is poorly understood what happens after WGD, how various rearrangements occur, and how much phenotypic impacts each rearrangement has") and novel findings ("We then argue potential evolutionary impacts of WGD allowing more complexed genome rearrangements and phenotypic diversification"). As soon as this reviewer read these two statements in the abstract, it was known that the authors were not familiar with the literature in this field. Perhaps the manuscript could be re-written with an emphasis on the novel method used to generate the materials; however the major findings are merely confirmatory in this reviewers opinion.

19th December, 2017

Dear Editor,
Associate Editor
Nature Communications

Muramoto et al. (NCOMMS-17-18885), Response to reviews

We appreciate your careful consideration and the reviewers' constructive comments, which helped strengthen and clarify the paper. We have addressed all of the points raised by referees by adding new data, modifying sentences and figures, and/or rebuttal. Our responses to their comments and suggestions are detailed below (reviewer comments in black text; responses in blue). We also indicated the positions of our revision in the revise manuscript with **yellow marker**.

Overview of major changes:

- In the revised manuscript, we reduced the emphasis on the aspects of whole genome duplication (WGD) and modified the entire manuscript to focus on the technological achievement of TAQing system, according to the reviewers' comments.
- We added new control experiments (viability in all possible conditions and experiments using a nuclease-dead mutant of TaqI). No conclusions were changed.
- We conducted analysis of 3D positioning of loci pairs for C-NHEJ-mediated translocations (TLs) using the previously published yeast Hi-C data. We additionally described an interesting tendency in the distance of loci pairs for TLs (see new Fig 2g-h).

Taken together, we now believe the manuscript has been substantially improved according to the reviewers' comments. We would be happy if you consider our manuscript suitable for the publication in *Nature Communications*.

To the Comments by Reviewer #1 (Remarks to the Author):

*In the manuscript "Monitoring the genome reorganization process after genome duplication revealed substantial phenotypic effects" the authors describe a system to generate chromosomal rearrangements by inducing genome-wide DNA double strand breaks (DSB) with the restriction endonuclease TaqI. The authors suggest that 'TAQing' is better tolerated in cells with higher ploidy, both in *S. cerevisiae* and *Arabidopsis*, and can produce phenotypic diversification. Additionally, the authors sequence genomes derived from TAQing to investigate the mechanisms*

by which DNA repair occurs following transient induction of genome-wide DSBs.

Overall I think that TAQing is a really interesting idea and a tool that could be broadly useful. For instance, TAQing could be used as a tool in metabolic engineering to generate new genomic sequences that improve pathway flux in microbial and plant production strains. Additionally, TAQing could be used as a tool to study aspects of genome chaos, gene networks, or potentially to generate minimal genomes in a completely unbiased manner and under different growth conditions.

Here the authors present TAQing primarily as a model for genome re-organization after a whole genome duplication event (WGD). Below I outline major and minor comments. With major revisions, in principle I am supportive of publication in Nature Communications and would be happy to review the manuscript again.

First of all, thank you for giving us constructive comments and suggestions.

Major comments

1. The title and abstract do not accurately reflect the contents of the manuscript. For instance the title doesn't mention that the genome reorganization process is achieved via induced double strand breaks. Further, there is no mention of TaqI in the abstract. Both the title and abstract should be modified extensively.

This point is well taken; we modified the title, abstract and introduction to reflect the contents more accurately.

2. Whether the simultaneous induction of ~200 DSB per cell at a defined sequence leading to genome rearrangement is an appropriate model for studying the downstream effects of WGD is a hard question to answer. The authors address shortcomings of their approach in the discussion.

We added the discussion point (shortcomings of TAQing system) as suggested in the discussion section (P.15, Line 287) as follows:

"In the present activation conditions, TAQing system is assumed to generate nearly 200 simultaneous DSBs, which may not be most suitable for the investigation of the DSB effects in cells with smaller ploidy. Usage of different restriction enzymes or activation conditions for TAQing may solve this problem."

3. More controls should be shown to establish the viability of yeast cells post TAQing. In Figure 1E, viability should be quantified for all conditions that could contribute to lethality and should include

(i) no Cu^{2+} /30°C (mock); (ii) Cu^{2+} /30°C; (iii) no Cu^{2+} /42°C; (iv) Cu^{2+} /42°C. Also, a *TaqI* endonuclease-dead mutant could be used to distinguish lethal genome rearrangements from lethality due to heat shock/copper.

To address this, we examined viability of yeast haploids as suggested (new Fig. 1e). As expected, we observed reduction of viability only in (iv) Cu^{2+} /42°C. In addition, we employed "TaqI endonuclease-dead mutant (dTaqI)" for additional control experiments. The viability of yeast cells expressing dTaqI stayed at high levels in any conditions. Furthermore, we repeated these experiments on minimum medium for the selection of TaqI-plasmid (SD/-Ura), according to the reviewer1's comment 5, and obtained almost the same data. Therefore, no conclusions were changed.

4. It isn't clear from the manuscript text or the figure legend whether the data in Figure 1E is from a haploid or diploid cell. In either case, viability should be quantified for both haploid and diploid cells. Presumably viability would be higher for the diploid cells and would provide additional support for the conclusion that TAQing is better tolerated by diploid cells.

We analyzed cell viability after TAQing in haploid, diploid, and tetraploid cells. We actually observed differences in viability of haploids and diploids, and added this new data (please see Fig 1f).

5. When quantifying viability, after TAQing cells should be plated on medium that selects for the TAQ plasmid rather than YPD. There is a huge selection pressure favoring cells that have lost the plasmid.

As mentioned in point-3, we repeated these experiments on SD/-Ura medium for the selection of TaqI-plasmid. No changes in conclusions.

6. For the yeast TAQing experiments, the authors should clearly state how many colonies post-TAQing were evaluated for genome rearrangements, by what method were they evaluated, and how many actually have rearrangements.

We added more detailed descriptions of this part on P.7 Line 133 and P.10 Line 189. To clarify this point, we also added a Supplementary Fig 2c describing the work flow as shown below.

7. Throughout the manuscript the authors refer to 'mock' versus TaqI activation. Please modify figure legends to clarify that the mock condition – for yeast I assume this means without Cu^{2+} at 30°C.

We modified the corresponding figure legends.

8. Why wasn't TAQing carried out in a tetraploid yeast cell? Isn't this a better model of WGD?

As mentioned in point-4, we analyzed the effects of TAQing system also in tetraploid yeast cells, but the difference in viability of diploids and tetraploids was little detected (please see the figure in Point-4). In addition, the degree of rearrangements in tetraploids was only slightly larger than that in diploids. It is thus possible that effects of WGD may be saturated in yeast at ploidy larger than diploid. We therefore focused on the difference in viability between yeast diploids and haploids in this study.

9. Lines 95-96 - it's not clear how the pulsed-field gel provides evidence to support this conclusion. Please clarify.

To clarify this point, we modified the text as follows on P.6 (Line 109):

"Judging from the degree of the broken DNA fragments of the PFGE analysis, we estimated that the DSB frequency was comparable to that in diploid meiotic yeast cells¹⁵, which reportedly produce ~200 DSBs per cell."

10. *In the discussion, the authors should reference and compare TAQing to SCRaMbLE (PMID: 265666580), which is another means by which to generate genome rearrangements in yeast.*

We added substantial description on the comparison with SCRaMbLE system in the discussion part (P.18 Line 349-).

11. *In principle, chromosomes that are close in nuclear space are more likely to undergo translocations as a function of TAQing. Have the authors attempted to correlate Hi-C data with TAQi induced genome rearrangements?*

We appreciate this reviewer's suggestion; we analyzed 3D positioning of loci pairs for the observed translocations (TLs) by referring with chromosomal contact information revealed by the previously published Hi-C analysis (Duan Z. et al., Nature 2010). As shown in Figure 2g and h, the average distance between chromosomal loci pairs for directly-joined unequal (C-NHEJ-mediated) TLs was smaller than the average distance of any random two chromosomal points. These data suggest that unequal TLs tend to occur between loci in relatively close proximity. We revised the text in P.9 Line 174-.

Minor comments

1. *Figure 1 legend – include the strain number for each experiment. It is difficult to tell whether each experiment is carried out in a haploid or diploid cell for panels (d) and (e).*

We included the strain number for each experiment in the legends of Figure1.

2. *Figure 1A – the homologous chromosomes should be the same size.*

Thank you; done.

3. *Line 36 – should read “we have established a new tool that induces DNA double strand breaks and extensively...”.*

We modified the text on P.2 (Line 34-).

4. *Line 105 – should read “diploid cells can undergo gross chromosomal rearrangements at a higher frequency than haploid cells.”*

We altered the text on P.7 (Line 128-).

5. Line 127 – define SGD

Thank you; done (P.8 Line 158).

6. Line 128-129 – isn't templated repair exactly the benefit of having a second chromosome following a DSB?

We modified the text of P.8 Line 159 according to your suggestion.

7. Line 129 – define TL

Done.(P.8 Line161)

8. Line 131 & 215 – I don't think the term microhomology is applicable here. Should be sticky end?

We used "cohesive overhang" on P.8 (Line 162), P14 (Line 264), and P.16 (Line 312).

9. Panel labels on the figures/figure layout are hard to follow. Should be rearranged so they appear in more logical order.

We apologize for the confusing panel layout. We have rearranged and revised the panel labels.

10. Line 171 – define GUS

Done. (P.11 Line 219)

11. The authors should include a strain table in the supplement with all parental strains as well as strain numbers for all analyzed strains in the manuscript (eg. strains produced by TAQing).

We added tables for the strains obtained in this study in Supplementary table S3 and S4.

12. Plasmids for TAQing should be contributed to Addgene.

Yes, we are now preparing for the plasmid contribution to Addgene.

To the Comments by Reviewer #2 (Remarks to the Author):

Understanding the mechanisms of chromosome rearrangements is an exciting and important area of research. It is well accepted that whole genome duplication (WGD) promotes large-scale genome rearrangements that provide raw materials for adaptive evolution. The problem, however, is that this idea relies on the genomic analysis of the past WGD-experienced species, and it remains poorly understood what exactly happens after WGD, how rearrangements occur, and whether these rearrangements can contribute to evolution. The study by Muramoto et al investigates chromosomal rearrangements induced in yeast and Arabidopsis following WGD. In particular, they established a very powerful and innovative tool to induce gross chromosomal rearrangements (GCRs). GCRs are induced by introducing simultaneous genomic DNA double strand breaks (DSBs), which are formed at one time in living cells by the heat-activated endonuclease Taq1. They applied this tool to yeast and Arabidopsis cells with different

ploidy, and concluded that yeast and plants with larger ploidy can undergo much more complex genome rearrangements. In addition, the authors report that that genome-rearranged yeast and plant cells showed enhanced phenotypic diversity. Overall, the authors propose that genome duplication allows more complex genome rearrangements and phenotypic changes that are promoting adaptive evolution. Overall, this new approach to the analysis of GCRs, their formation, and connection with WGD and evolution represents an important development in the field of genetic stability and evolution, and the results of this study are of interest for researchers working in many areas, including DNA repair, recombination, and evolution. The side-by-side comparison of yeast and plants and of organisms with different ploidy combined with detailed characterization of mutations and genome rearrangements genome-wide are among the main strengths of this work. Because interest in this topic is very high, I expect that this paper will be frequently cited and this approach will be used in various areas.

Thank you for positive comments and helpful suggestions.

Specific comments:

1. P.2. Abstract.

“ We found large-scale rearrangements occur frequently in diploid yeasts and tetraploid plants, whereas haploid yeasts and diploid plants were little rearranged”. This is the most important conclusion of the paper, and it is definitely supported by data and exciting. However, it is important to compare, side-by-side, cell viability and the number of breaks produced by Taq1 in yeast haploids vs diploids and in plants. I was able to find (Fig. 1e) that Taq1 induction leads to the lethality of at least 40% of cells in haploids. Is it possible that all rearrangements occurred in this fraction? How does it compare with viability in diploids? The lethality of GCRs in haploids is the simplest explanation of the main result, and it will be important to provide all of the necessary comparisons. Also, is it that the number of breaks in diploids is approximately 2x of that in haploids?

Your thoughts are well taken. To address this, we analyzed lethality in haploid, diploid, and tetraploid cells and added new data for the comparison of lethality of TAQing on haploids and diploids (Fig. 1f). Lethality of diploid cells after TAQing was lower than that of haploid, though the number of DSBs are assumed to be equivalent in haploids and diploids. More importantly, the survived haploid cells did not have any PFGE-visible gross rearrangements (Supplementary Fig. 2a, b). These results indicate that the TAQing system can induce large-scale genome rearrangements much more efficiently in yeast diploids than in haploids.

Possibly, haploids are less tolerant for gross genome rearrangements (namely "the lethality of GCRs in haploids" that you suggested) than diploids, due to their lower redundancy of genetic information. We described these points in the result section.

2. Fig. 2g and p. 5, line 101. *"In addition, heat activation markedly induced gross chromosomal rearrangements". Fig. 2g represents loss of heterozygosity events. Why are they chromosomal rearrangements if they result from exchange between homologs?*

We apologize for the ambiguous description. (We think "Fig. 2g" might be "Fig. 1g", judging from the context.) We revised the text to improve the clarity on P.6 (Line 119-). We also modified the figure legend of Fig. 1h (previously Fig 1g).

3. P. 6, line 111. *" Genomic alterations induced by activated Taq1 were further examined using strains with altered morphology ("mutants") by combined long/short high throughput sequencing". From this statement it appears that the mutants were selected. It will be helpful to describe the process of selection further. In particular, it is important to explain how the frequencies of chromosomal rearrangements observed in this experiment characterize the frequency of rearrangements in the entire population.*

According to the suggestion, we added more detailed descriptions of this part on P.7 Line 133- and P.10 Line 189-. We also incorporate a Supplementary Fig 2c describing the work flow and lists of the strains obtained in this study (Supplementary table S3 and S4).

4. P. 6, line 120. *"These mutations likely resulted from error-prone DSB repair consistent with a previous report". The authors propose that mutations result from the mechanism that was proposed to result from the damage accumulated in ssDNA resulting from the DSB resection. Do the authors believe that these mutations were associated with repair of Taq-induced breaks that proceeded not via NHEJ, but via HR or MMEJ associated with long resection? Some discussion will be very useful here.*

To clarify these points, we argued the possible mechanisms (C-NHEJ, HR, A-NHEJ/MMEJ) for the mutations induced by TAQing on P.8 (Line 151-):

"It is therefore suggested that these mutations, as reported previously, were at least partly induced by error-prone DNA repair of TaqI-mediated DSBs that proceeded not via canonical nonhomologous endjoining (C-NHEJ), but via HR or microhomology-mediated end joining (MMEJ) (or alternative NHEJ, A-NHEJ) associated with rather long resection."

5. P. 7, line 127. *The term SGC needs explanation. I was able to find its meaning only in figure legend.*

Thank you; done.

6. P. 7, line 128. *"This is surprising since homologues are assumed to exist independently in vegetative yeast nuclei". I am not sure why it is surprising. Gene conversion, especially allelic gene conversion is very efficient in yeast and does not require homologues to stay together all the time.*

We agree with the reviewer's comment and altered this part on P.8 (Line 158) by simply describing:

"We detected 40 non-reciprocal inter-homologue short gene conversions (SGCs, 0.7–30 kb, average 7.0 kb in length) (Fig. 2a, b), supporting that yeast cells undergo HR-based repair very efficiently."

7. P. 7, line 129. *"Detectable TL breakpoints were observed in 66.6% (6/9) of Taq1 recognition sequences". This requires clarification. Do the authors mean all Taq1 recognition sequences around the genome? Or do they mean that every breakpoint correlated with a Taq1 recognition sequence?*

We are sorry for the confusing description. The latter is the case. We clarified this point by describing:

"All six nonhomologous TL breakpoints were on Taq1 recognition sequences." (P.8, Line 160).

8. P. 7, line 135. *"Two isolates harbored chromosomes with fused TL breakpoints with Ty elements, and one isolate had a chromosome fusion with a repetitive rDNA region. This is intriguing, as a previous study suggested that TLs inoriginate from HR between Ty elements or repetitive sequences". Could the authors clarify their statement: "this is intriguing"? Do they want to say that their finding contradicts those previous reports? It looks like they found the same thing. Could they please explain or re-formulate?*

We intended to state that our findings were consistent with previous studies. We modified this part to clarify this point (P.9, Line 169).

9. P. 36, Fig. 2d. How was mutation rate calculated? Was it calculated per generation? How many times was the experiment repeated? How were statistics for this experiments done?

We apologize for the ambiguity of "mutation rate". This part should be "the frequency of canavanine-resistant colonies" which reflects mutation in *CAN1* gene. The experiment was repeated for 4 times independently. The statistics was analyzed by Welch t-test (*: $p < 0.05$). To describe these more accurately, we revised the text (P8. Line 148) and figure legend for Figure2c.

10. P. 36 Fig.2e legend: "mutated chromosome": does it mean "rearranged chromosome"?

Thank you; done.

To the Comments by Reviewer #3 (Remarks to the Author):

The authors of "Monitoring the genome reorganization process after genome duplication revealed substantial phenotypic effects" describe a method where they induce double strand breaks given an enzyme. Furthermore, they investigate the result of these induced breaks and measure the influence of a whole genome duplication effect. The introduction, result and discussions are well written and easy to follow. They successfully test the activation of their enzyme using a heat response and quantify translocations and duplication events between different mutants and the mock cell line.

We appreciate your positive and helpful comments.

In the following I list my concerns and questions about specific topics:

1. Some sections especially in the Methods are not well written and thus hard to understand. I would advise to check and correct the text.

We apologize for confusions and ambiguity in the Methods section. We thoroughly revised the text and added Supplementary Fig 11, 12, 15 and 16 to clarify the process of data analysis.

2. Why does the *Taq1* produce more stable genotypes? As far as I understood it is a cutter enzyme. Especially for the yeast data set you see only a fraction of the cells show some phenotypic changes, right? Have you assessed other methods to induce DSB? Furthermore, I had troubles to identify the method with what the mock samples were treated.

About stable genotype inheritance, in this study we intended to describe stable transmission of the combination of multiple traits. During multiple rounds of passages, some beneficial combination of mutant alleles may be lost due to chromosome-wide loss of heterozygosity. TAQing enables rearrangements that enable linkage of multiple mutant alleles on the same chromosome which can be more stably transmitted to daughter cells.

About the fraction with phenotypic changes, we think many of TAQed-cells can have different types of phenotypic changes, which may not be always detectable by limited ways of phenotype detection.

About other DSB inducing methods, we indeed tested other DNA endonucleases. But the results are beyond scope of this paper. We also believe that TAQing system has the advantage of easy estimation of potential nearest DSB sites that induced the chromosomal rearrangements. We described this advantage of TAQing system on P5. (Line 98):

"This TCGA sequence can help us to consider the causal relationship between rearrangement events and the adjacent potential DSB sites, which are difficult to estimate in radiation- or mutagen-based conventional mutagenesis."

We defined "mock treated samples" more precisely ("Mock; no Cu²⁺/30°C" in legends for Fig1f).

3. You mention for *Arabidopsis* that the phenotype changes (leave sizes) were successfully transmitted, but leave out at what rate or over how many generations. I think that is important for your claim that changes induced are stable over the inheritance and especially "over many generations". Also for the yeast data the replication numbers that support your claim are not high and thus not 100% conclusive.

We agree with your opinion and altered the corresponding description. For example, in yeast test of Fig3e; "at least after 10 passages", and plant test in Fig5a; "over at least three generations".

4. Please be more precise in e.g. "some isolates exhibited different..." (Page 8)

We described the number of isolates more precisely on P.10 (line 190).

5. You need to improve the explanation of the computational methods and parameters used for the sequence analysis. For example, I had the impression that the analysis was different for the yeast vs. *Arabidopsis* samples. If this is the case, then please make it clear. When did you use

which genome (e.g. multiple yeast strains were assembled)? You are also sometimes mentioning that you used CLC and sometimes you seemed to use more direct methods e.g. BWA. I would furthermore recommend to cluster the subparts of the methods section according to field. E.g. all computational method description should be grouped together to enable an easier understanding of what was done.

We used different NGS data analysis pipelines for yeast and plant. BWA was adopted for mapping, and GATK-tool was employed for variant detection in yeast. For the study of plant, mapping and variant detection was executed by CLC workbench. In addition, we described on P13 "We then conducted combined long-read/short-read high-throughput resequencing of TAQed plants". This is our mistake. We indeed conducted long-read analysis of plants but did not use the data in the present analysis. We therefore altered this part to "We then conducted combined analysis using short-read high-throughput resequencing and tiling-array analysis of TAQed plants". We described these in the methods section, as suggested. Descriptions on computational analyses were clustered in the last part of Methods section. We also refined the methods section of computational analysis and added the Supplementary figure 11–16 for easier understanding.

6. Please consider rewriting the section "Validation of structural variants". I don't understand "accumulated unaligned reads (>2 reads) on the boundary line of CNV". How can you observe that these reads were unaligned, but belong to a certain region? Furthermore, 2 reads are not very stringed filter. Did you carry out any additional analysis for smaller variations or translocations?

Yes, we did. We extensively rewrote the section "Validation of structural variants" and added the supplemental figure for explaining the three criteria of validation (Supplementary Figure 16).

7. For the translocations that you observe do you have validation data? I don't see that you mention this. Especially these translocations between repetitive elements are often error prone depending on the analysis.

These TL sites except for TLs between rDNA regions were physically verified by PCR amplification using the adjacent DNA sequences of the TL junctions. We mentioned this point in Methods section (P.33, Line 627). The TL within rDNA is indeed hard to confirm, but the changes in read depth of rDNA region clearly support that the TL occurs within rDNA.

8. I could not find the assembly statistics to see the “high quality reference genome assembly”. What does “highly consistent with previous reported S288C genome sequence” mean? I would have liked to see some numbers of SNPS, alignment statistic, etc. Furthermore, you are writing “N50 coting’s were partially corrected”. I don’t know what this means. N50 is a size measure, but you don’t give any numbers.

We apologize for confusions in this part. We added information about SNP and InDel frequencies in Supplementary figures:

1. Information about frequencies and distribution of SNP and InDel between S288C-YPH499 and YPH499-S799; Supplementary Figure 13 and 14.
2. Information about alignment statics; Supplementary Table S5.

9. For the “Mutation detection analysis” What method was used to detect the mutations? What method did you use to identify CNVs? I am a bit confused as to what you refer to translocations and CNVs. These are two or more completely different types. Translocations are most often referred as connections between two chromosomes. Whereas CNVs are alterations in the coverage (most often duplications and deletions). It can be that some translocations explain e.g. duplications for example in cancer. But they are still different types. Please clarify: “Chromosomal rearrangements for which the CNVs were also confirmed by the tiling array data were defined as “translocation,” and the corresponding mutation points were detected by extracting the chimeric reads around the junction sites” (Page 25).

We carefully revised the Methods section for “Mutation and rearrangement detection analysis”. In addition, we clarified the text on P.32 (Line 616): “Chromosome wide CNVs detected by both sequencing and tiling array were defined as “aneuploidy”. Chromosomal rearrangements with the junction of different segments of chromosomes were defined as “translocation (TL)...”. For easier understanding the Method section for variant detection, we added a new supplemental figure 15.

10. Some acronyms are hard to find (e.g. only mentioned in the figure legend: SGC)

We carefully reworked the definitions of all acronyms including short gene conversions (SGC), β -glucuronidase (GUS) and translocaltion (TL).

To the Comments by Reviewer #4 (Remarks to the Author):

The main claim of the paper is that it is poorly understood what happens immediately after polyploidization with respect to genome reorganization and its potential impact on phenotype. This reviewer agrees with the authors that it is important to do experimental evolution where one starts with a diploid plant (or haploid yeast) and then induce various mutations (SNPs, translocations, aneuploidy, auto-polyploidy, and allopolyploidy) to see what impact these would have on phenotype. The authors have also developed a novel method to generate genome structural variation that works in plants and yeasts. However, this reviewer has two major concerns with the paper.

*First, it is false for the authors to claim novelty that they are the first to perform experimental evolution to make and watch the consequences of polyploidy from known parents. Second, the major conclusion that polyploids have more evolutionary space for genome rearrangements and these have a major impact on phenotype is not novel; indeed, it is absolutely expected and it would be a surprise at this point to find the opposite. The authors are simply unaware of this literature (or chose not to cite it in this paper). Just three references from one species (*Brassica napus*, or canola) are given here:*

**Gaeta et al. 2007. Genomic changes in resynthesized *Brassica napus*: The effect of genomic changes on gene expression and phenotypic variation. *The Plant Cell* 19: 3403-3417*

**Xiong et al. 2011. Homoeologous shuffling and chromosome compensation maintain genome balance in resynthesized allopolyploid *Brassica napus*. *Proceedings of the National Academy of Sciences, U.S.A.* 108: 7908-7913.*

**Chalhoub et al. 2014. 2014. Early allopolyploid evolution in the post-neolithic *Brassica napus* oilseed genome. *Science* 345: 950-953.*

The last paper includes both natural and resynthesized polyploids.

*One can also find similar works on resynthesized polyploids and their major rearrangements and effects on phenotype in the literature in *Arabidopsis* (Luca Comai, Jeff Chen, Andreas Madlung labs), maize (Jim Birchler lab), wheat (Bao Liu's lab) and *Tragopogon* (Pam and Doug Soltis lab). Similarly, there has been a lot of work done in resynthesized yeast polyploids (Sally Otto and other labs). So it is no surprise the authors find their approach and major finding novel if they are not familiar with this vast body of literature.*

Second, the authors methodology for examining genome reorganization is flawed in that they do not distinguish gains and losses from homoeologous exchanges which may do both

simultaneously (Figures 5b, 6b, 6c, Supplemental Figures 7b, 8, 9b, 10). In the yeast case some of these may be correctly referred to as gene conversions; however, in the tetraploid plant case this may not be so. There is also the possibility of hidden aneuploidy (where one homoeologous chromosome is completely lost and replaced by its partner, although the results do not seem to show that). Some of the references given above perform these analyses and discuss these points.

In summary, this reviewer finds this manuscript deeply flawed with respect to making claims about novel experimental design (“it is poorly understood what happens after WGD, how various rearrangements occur, and how much phenotypic impacts each rearrangement has”) and novel findings (“We then argue potential evolutionary impacts of WGD allowing more complexed genome rearrangements and phenotypic diversification”). As soon as this reviewer read these two statements in the abstract, it was known that the authors were not familiar with the literature in this field. Perhaps the manuscript could be re-written with an emphasis on the novel method used to generate the materials; however the major findings are merely confirmatory in this reviewers opinion.

We sincerely apologize for causing misunderstanding by the reviewer #4 about the essence of this paper. Please understand that we never wanted to underestimate the previous works. Main achievement of this study is a development of new system to monitor the genome rearrangements and its impacts on phenotypes after induction of multiple simultaneous DNA breaks. As suggested, WGD effects have been extensively investigated previously (we cited some of the suggested papers in the revised manuscript), but the effects of WGD in combination with multiple DSB formation have not been demonstrated. To clarify this, in the revised manuscript, we reduced the emphasis on the aspects of WGD and modified the entire manuscript to focus on the technological achievement of TAQing system.

We are also sorry for our confusing description on our genomic sequence analysis. Our analysis practically enabled identification of gene conversion events by detecting SNP-based loss of heterozygosity. In yeast, reciprocal crossing-over can be detected by the generation of chimeric junction sequences in long-read/short-read analysis. We also confirmed the differences in copy number by tiling array experiments. We carefully rewrote Methods section to clarify this point.

Reviewers' comments:

Reviewer #1 (Remarks to the Author):

The authors have attempted to address most of the concerns of all four reviewers and the manuscript is much improved. Additional clarification is required for Figure 3E and accompanying supplementary information and the authors should re-write/clarify this section. With these changes, I support publication of this manuscript.

Major comments:

1. The abstract and title are much improved, however, the last sentence of the abstract is unclear and should either be removed or rewritten. The abstract should be carefully copy edited.
2. Lines 137-138 – Since the authors pre-selected for phenotypic changes and then went looking for genome rearrangements the conclusion should be that genome rearrangements frequently accompany phenotypic changes, not the other way around as it currently written.
3. Line 141 – the mating type of WT14, which is a product of fusion of YPH499 and S799, is listed as MATa/a. In the supplementary yeast strain table, S799 is listed as a MATa. Is this a typo in the strain table?
4. Line 188 – change to “TAQed yeast genomes confer marked phenotypic diversity”
5. Line 189-195 – this whole section is a circular argument – the authors pre-selected colonies for further study based on reduced colony size and alteration in cell morphology following TAQing. Then they showed that genome rearrangements are found at high frequency in these TAQed strains. They can't go back now and conclude that phenotypic diversity accompanies genome rearrangements. This text should be deleted or moved up to section “TAQing-induced chromosome rearrangements often occur between homologous sequences and unprocessed DSB end” and presented as a characterization of types of phenotypes that arise from TAQing.
6. Figure 3E, Supplementary Figure 5, and related manuscript text – there is simply not enough information to understand the conclusions. Specifically:
 - a. Line 201 – more details should be included on strains used for growth curves In Figure 3E. Were single colonies post-TAQing chosen for growth curves? How many? Were they cultured independently or together?
 - b. Figure 3E – what is the control strain? Shouldn't this be the fused diploid pre-TAQing? If I have guessed correctly, shouldn't this strain be able to grow on the two individual selective conditions? If this isn't the strain, then it should be included in this figure and the 'control strain' should be defined.
 - c. Supplementary Figure 5A – I don't understand why none of the TAQed strains (CF1-6) share the high temp and xylose growth phenotypes? Presumably one (or more) of these strains was used for the growth curve in Figure 3E?
 - d. Lines 204- 206 – it's not clear to me how the data presented in Supplementary Figure 5 support this statement. Unless CF1-CF6 are fusion strains pre-TAQing?
 - e. Line 208/Supplementary Figure 5B – indicate number of generations rather than number of sub-culturing steps. More details should be included about how the passaging was carried out.
7. The manuscript needs copy editing throughout.

Minor comments:

Figure 3 – typo in figure legend: diversification

Table S1 – define acronyms (SNP, SGC, BIR, etc)

Supplementary Figure 2C – should be " $<4/5$ in area compared to the control"

Reviewer #2 (Remarks to the Author):

The authors addressed the reviewer's comments really well. The manuscript is very much improved in my opinion. I have only really minor comments regarding the new text.

1. p.6, line 115. " When a 114 nuclease-dead mutant TaqI (dTaqI with K142A mutation) was introduced to yeast cells, such
115 effects on viability could not observed at all (Fig. 1e), indicating that the reduced viability is indeed caused by the TaqI DNA cleavage reaction"
This sentence needs to be re-phrased. Probably "could not" needs to be replaced, or smth. else changed since it does not make sense the way it stands.

2. p. 6, line 121
"that TaqI heat activation markedly induced
121 rearrangements such as gene conversions or unequal crossing overs".
Gene conversion is not rearrangement.

3. p.6, line 122.
"It should be noted that 28.1% (9/32) of activated TaqI-treated ("TAQed") diploids
123 exhibited changes in chromosome size, while only 3.1% (1/32) of mock controls in PFGE.
"In PFGE" seems to be not in place. It is unclear this way.

4. p. 7, lines 133 -137.
"we selected 178 mutants....Large portion of such mutant
isolates (11/13) exhibit chromosomal size alteration detectable in PFGE analysis (Fig. 1i)."
So, it is unclear how many mutants were analyzed by PFGE: 178 or 13?

Reviewer #3 (Remarks to the Author):

The authors have addressed all my concerns and improved the description of the method section where it was needed.

8th March 2018

Dear Editor,

Associate Editor

Muramoto et al. (NCOMMS-17-18885), Response to reviews

Thank you very much for careful and kind considerations on our paper. We have responded to all comments by the reviewers.

We would like to thank all the reviewers for their elaboration and giving us many valuable suggestions to improve our paper.

Taken together, we now believe the manuscript has been fully responded to the reviewers' comments. We would be happy if you consider our manuscript suitable for the publication in *Nature Communications*.

Our responses to their comments and suggestions are detailed below (reviewer comments in black text; responses in blue). We also indicated the positions of our revision in the revised manuscript with **yellow marker** and copy-edited parts are indicated with **shadowing marker**.

In addition, we eliminated a table from Fig. 2 and newly prepared Table 1 to be fit with the Nature Comm. format. And the uncropped blots for Fig. 1b and c were added in Supplementary Fig. 11 with mentioning on P.26 Lines 490-491.

Reviewers' comments:

Reviewer #1 (Remarks to the Author):

The authors have attempted to address most of the concerns of all four reviewers and the manuscript is much improved. Additional clarification is required for Figure 3E and accompanying supplementary information and the authors should re-write/clarify this section. With these changes, I support publication of this manuscript.

Major comments:

1. The abstract and title are much improved, however, the last sentence of the abstract is unclear and should either be removed or rewritten. The abstract should be carefully copy edited.

According to the reviewer's suggestion, the title and the last sentence in the abstract was copy edited and rewritten as follows:

Title: Phenotypic diversification by enhanced genome restructuring after induction of multiple DNA double strand breaks

" This genome-restructuring system (TAQing system) will enable rapid genome breeding and aid genome-evolution studies." (P.2, Lines 38-39)

2. Lines 137-138 – Since the authors pre-selected for phenotypic changes and then went looking for genome rearrangements the conclusion should be that genome rearrangements frequently accompany phenotypic changes, not the other way around as it currently written.

We agree the reviewer's opinion and altered the problematic statement as follow:
"suggesting that the massive genome rearrangements frequently accompanied phenotypic changes." (P.7, Lines 128-129)

3. Line 141 – the mating type of WT14, which is a product of fusion of YPH499 and S799, is listed as MATa/a. In the supplementary yeast strain table, S799 is listed as a MATa. Is this a typo in the strain table?

Thank you for letting us this typological error. We modified it in supplementary table 4 and the methods section as follows. "Yeast strains YPH499 (S288c-derived haploid; *MATa_ura3-52 lys2-801 ade2-101 trp1-Δ63 his3-Δ200 leu2-Δ1*) and S799 (SK1-derived haploid; *MATa_ura3 lys2 ho::LYS2 leu2Δ arg4-bgl cyh2-z*)" (p.20, Lines 367-368)

4. Line 188 – change to "TAQed yeast genomes confer marked phenotypic diversity"

Thank you. We altered the sentence as suggested (P.10, Line 179 subheading).

5. Line 189-195 – this whole section is a circular argument – the authors pre-selected colonies for further study based on reduced colony size and alteration in cell morphology following TAQing. Then they showed that genome rearrangements are found at high frequency in these TAQed strains. They can't go back now and conclude that phenotypic diversity accompanies genome rearrangements. This text should be deleted or moved up to section "TAQing-induced chromosome rearrangements often occur between homologous sequences and unprocessed DSB end" and presented as a characterization of types of phenotypes that arise from TAQing.

To avoid the circular argument, we deleted the problematic statements and rewritten this part as follow:

"In addition to variability in the cell size and morphology (see Fig. 3a; Supplementary Fig. 2d), we also detected substantial non-morphological qualitative changes in the TAQed yeast diploids. For example, 6.2% (11/178) isolates exhibited hyper- or hypo-flocculation phenotypes (Fig. 3b and c; Supplementary Fig. 2c). This TAQing-induced phenotypic diversity led us to further examine whether different traits derived from two strains could be stably combined, as the two strains are subjected to cell fusion, followed by TAQing treatment." (P.10 Lines 180-185).

(Additionally, we found Sup.Fig2d and Fig3d are somehow redundant. We therefore deleted Fig3d.)

6. Figure 3E, Supplementary Figure 5, and related manuscript text – there is simply not enough information to understand the conclusions. Specifically:

a. Line 201 – more details should be included on strains used for growth curves In Figure 3E. Were single colonies post-TAQing chosen for growth curves? How many? Were they cultured independently or together?

We added more detailed descriptions as follow:

"We then heat treated the freshly prepared fused non-TAQed diploid to induce TAQing, followed by single colony isolation on xylose-containing medium at 40°C. As a control, non-TAQed fused strains were subjected to the same procedure in the absence of TAQing (Supplementary Fig. 5a)."

(P.10 Lines 191-194).

b. Figure 3E – what is the control strain? Shouldn't this be the fused diploid pre-TAQing? If I have guessed correctly, shouldn't this strain be able to grow on the two individual selective conditions?

If this isn't the strain, then it should be included in this figure and the 'control strain' should be defined.

Yes, this is the pre-TAQing fused diploid.

We explained this strain in the text (see above) and with a diagram in Suppl.Fig 5a.

We also modified the legend for Fig 3e.

c. Supplementary Figure 5A – I don't understand why none of the TAQed strains (CF1-6) share the high temp and xylose growth phenotypes? Presumably one (or more) of these strains was used for the growth curve in Figure 3E?

We added the following explanation:

" To investigate this, we fused a xylose-utilizable haploid strain (capable of growing on xylose-containing medium) and a thermotolerant haploid strain (which can grow at 40°C) to obtain a non-TAQed diploid cell-fusion strain. This fused non-TAQed strain initially exhibited both xylose-fermentation ability and thermotolerance; however, after multiple passages, one of such combined phenotypes was lost, as it ceased to grow on xylose-containing medium at 40°C." (P.10 Line 186-190).

"Notably, only the TAQed diploid continued to grow stably under the combined stress environment during selection (Fig. 3d), suggesting that the non-TAQed fusion lost one of the combined traits during preculture." (P.11 Lines 195-197).

d. Lines 204- 206 – it's not clear to me how the data presented in Supplementary Figure 5 support this statement. Unless CF1-CF6 are fusion strains pre-TAQing?

As the reviewer expected, CF1-6 are all non-TAQed strains. As mentioned above, the fused strain without TAQing treatment easily lost one of the favorable phenotypes (Supplementary Fig. 5b-d).

e. Line 208/Supplementary Figure 5B – indicate number of generations rather than number of sub-culturing steps. More details should be included about how the passaging was carried out.

As mentioned above, we added a diagram of passages in Suppl.Fig 5a.

7. The manuscript needs copy editing throughout.

The manuscript is now checked by a professional copyediting service.

Minor comments:

Figure 3 – typo in figure legend: diversification

Thank you, we modified.

Table S1 – define acronyms (SNP, SGC, BIR, etc)

We define these acronyms in the revised manuscript.

Supplementary Figure 2C – should be "<4/5 in area compared to the control"

We are sorry for confusing description of this part. This is not <4/5 in area, but <1/5 in area. To avoid misunderstanding, we modified the corresponding description (P.7, Line 125) to "we selected 178 mutants which formed smaller colonies (<20% of the area)."

Reviewer #2 (Remarks to the Author):

The authors addressed the reviewer's comments really well. The manuscript is very much improved in my opinion. I have only really minor comments regarding the new text.

1. p.6, line 115. "When a nuclease-dead mutant TaqI (dTaqI with K142A mutation) was introduced to yeast cells, such effects on viability could not be observed at all (Fig. 1e), indicating that the reduced viability is indeed caused by the TaqI DNA cleavage reaction"

This sentence needs to be re-phrased. Probably "could not" needs to be replaced, or smth. else changed since it does not make sense the way it stands.

Thank you. The sentence was rewritten as "Introduction of an inactive TaqI variant (dTaqI; with a D142A mutation) into yeast cells resulted in no such effects on viability (Fig. 1e), indicating that the reduced viability was caused by TaqI DNA cleavage." (P.6, Lines 105-107)

2. p. 6, line 121, "that TaqI heat activation markedly induced rearrangements such as gene conversions or unequal crossing overs". Gene conversion is not rearrangement.

According to the reviewer's suggestion, we simply mention as follows:

"In addition, loss-of-heterozygosity assays (Fig. 1h) showed that TaqI heat activation markedly induced gene conversion or unequal crossing over." (P. 6, Lines 110-112)

3. p.6, line 122. "It should be noted that 28.1% (9/32) of activated TaqI-treated ("TAQed") diploids exhibited changes in chromosome size, while only 3.1% (1/32) of mock controls in PFGE. "In PFGE" seems to be not in place. It is unclear this way.

We changed this part as follows:

"PFGE experiments revealed that 28.1% (9/32) of activated TaqI-treated ("TAQed") diploids exhibited changes in chromosome size, whereas this was observed in only 3.1% (1/32) of mock controls." (P.6, Lines 113-115)

4. p. 7, lines 133 -137. "we selected 178 mutants....Large portion of such mutant isolates (11/13) exhibit chromosomal size alteration detectable in PFGE analysis (Fig. 1i)." So, it is unclear how many mutants were analyzed by PFGE: 178 or 13?

We added number of the mutants analyzed:

"11/13 mutant isolates exhibited altered chromosomal size detectable by PFGE analysis (Fig. 1i)," (P. 7, Lines 126-127)

REVIEWERS' COMMENTS:

Reviewer #1 (Remarks to the Author):

The authors have addressed all of my concerns.